# SARAH-3 – satellite-based climate data records of surface solar radiation

Uwe Pfeifroth, Jaqueline Drücke, Steffen Kothe, Jörg Trentmann, Marc Schröder and Rainer Hollmann

Deutscher Wetterdienst, satellite-based climate monitoring, Offenbach, Germany

Correspondence: Uwe Pfeifroth (uwe.pfeifroth@dwd.de)

**Abstract**

The amount of energy reaching the Earth's surface from the sun is a quantity of high importance for the climate system and for renewable energy applications. SARAH-3 is a new edition of a satellite-based climate data record of surface solar radiation parameters, generated and distributed by the European Organisation of Meteorological Satellites (EUMETSAT) Satellite Application Facility on Climate Monitoring (CM SAF). SARAH-3 provides data from 1983 onwards, i.e. more than 4 decades of data; provided with a spatial resolution of 0.05° x 0.05° and a temporal resolution of 30-minutes, daily means and monthly means for the region covered by the METEOSAT field of view (65°W to 65°E and 65°S to 65°N). SARAH-3 consists of seven parameters: surface irradiance, direct irradiance, direct normal irradiance, sunshine duration, daylight, photosynthetic active radiation and effective cloud albedo. SARAH-3 data between 1983 and 2020 have been generated with stable input data (i.e. satellite and auxiliary data) to ensure a high temporal stability; these data are temporally extended by an operational near-real time processing – the so-called Interim Climate Data Record. The data record is suitable for various applications, from climate monitoring to renewable energy. The validation of SARAH-3 shows a good accuracy (deviations of ~5 W/m$^2$ from surface reference measurements for monthly surface irradiance) and stability of the data record and further improves over its predecessor, SARAH-2.1. One reason for this improved quality is the new treatment of snow-covered surfaces in the algorithm, reducing the misclassification of snow as clouds. The SARAH-3 data record reveals an increase of the surface irradiance (~ +3W/m$^2$/decade) during the last decades in Europe, in line with surface observations.

## 1 Introduction

Surface solar radiation is of high importance for the Earth's climate (Ramanathan et al. 2001, Wild et al., 2012) and for life on Earth in general. Beside the astronomical Earth-Sun constellation and the individual daytime and location, surface solar radiation (SSR) is controlled by the atmospheric and surface properties. Overall, an important factor influencing SSR are clouds, which strongly reflect solar radiation / reduce SSR and are highly variable in space and time (Pfeifroth et al., 2018a, 2018b, Wild, 2012, Hartmann et al., 1986). Hence a dense observational network is required to capture the temporal and spatial variability of SSR. However, station-based high quality SSR measurements are often available at relatively few stations, e.g. from the Baseline Surface Radiation Network (BSRN), which do not capture neither the global nor regional SSR spatial and temporal distributions appropriately. Large gaps in space and time exist in the surface network, especially over the ocean and on the African continent.

Satellite data have become a valuable data source to fill the gaps (e.g. Gautier et al., 1980, Pinker et al., 1992, Huang et al., 2019) – not only in space, but also in time. SSR has been estimated from satellite measurements since the 1980s using a range of different retrieval methods (Rigollier et al., 2004; Vernay et al., 2004; Möser and Raschke, 1984; Cano et al., 1986; Müller et al., 2015b, 2022). The generation of longer-term data records, however, has only been started in the 2000s, when higher quality satellite data became available for one decade or longer. The data used for the monitoring of the climate typically is required to cover multiple decades (i.e., 20 years or more) and to be temporally homogeneous, in addition to have a high accuracy. For a comprehensive review of available retrieval techniques and selected data records as well as future perspectives the reader is referred to Huang et al., 2019.

Here we are presenting the climate data record (CDR) SARAH-3 (Pfeifroth et al., 2023, https://doi.org/10.5676/EUM_SAF_CM/SARAH/V003) generated by EUMETSAT's Climate Monitoring Satellite Application Facility (CM SAF, Schulz et al., 2009), i.e., the latest version of the series of SARAH CDRs. SARAH-3 has been released in May 2023 and covers more than 40 years (1983 to date), including, for the first time, the current WMO climate normal period: 1991-2020. SARAH-3 provides seven surface solar radiation parameters: solar irradiance (also called global radiation), two direct irradiance parameters (horizontal and normal), sunshine duration, two spectral surface radiation parameter (i.e. PAR, DAL) and the effective cloud albedo.

SARAH stands for 'SurfAce Radiation DAtaset Heliosat'. The data are based on the series of the geostationary METEOSAT Satellites of the first and second generation. The first METEOSAT-based SSR data record has been released by CM SAF more than a decade ago (Posselt et al, 2011) and with its successors SARAH-1, SARAH-2 and SARAH-2.1 the generated data have been steadily improved and extended in time. While for SARAH-1, the main step was the inclusions of the MVIRI sensor (onboard the 1$^{st}$ METEOSAT generation) and the SEVIRI sensor (onboard the 2$^{nd}$ METEOSAT generation) (Müller et al., 2015b), the stability over time was further improved with SARAH-2 (covering 1983-2015). SARAH-2.1 is the extension of the SARAH-2 CDR and came with a near-realtime processing for the first time. The so-called Interim Climate Data Records (ICDR) operationally andconsistently extended the SARAH-2 CDR with a delay of 2-3 days. The current SARAH edition, SARAH-3, is also accompanied and temporally extended by ICDR data, which enables climate monitoring applications (e.g. C3S, 2023). The main conceptional improvement in the generation of SARAH-3 has been the improved estimation of the surface solar radiation parameters in case of snow-covered surfaces, which reduced the underestimation of surface solar radiation and sunshine duration found in previous editions of SARAH (e.g., Niermann et al., 2019). The estimation of surface irradiance under snow-covered conditions has been identified as a key difficulty also for other retrieval techniques (see Huang et al., 2019). Two novel parameters (compared to previous SARAH editions), representing different spectral information, are included in SARAH-3, namely Daylight (DAL) and Photosynthetic Active Radiation (PAR). Some more information on the new parameters are given in section 2.

All CM SAF data records are freely available via the CM SAF Web User Interface (see www.cmsaf.eu, wui.cmsaf.eu) in NetCDF-format. The quality of the previous editions of the SARAH climate data records has been externally assessed under different aspects and for different regions by numerous studies, e.g., Urraca et al., 2017, Montero-Martin et al., 2020, Young and Bright, 2020; Mabasa et al., 2021; Kenny and Fiedler, 2022; Gava et al., 2023; Ouhechou et al., 2023; Sawadogo et al., 2023; Forstinger et al., 2023.

Applications of the available SARAH climate data records cover many fields and applications, including climate analysis and monitoring (e.g. Pfeifroth et al., 2018a, 2018b; Cebulska and Kholiavchuk, 2022; Obregon et al., 2014; C3S, 2023), evaluation of numerical models (e.g. Alexandri et al., 2015; Chen et al., 2024), agrometeorology (e.g. Pelosi et al., 2022), as well as analysis of surface station locations and quality control of surface data (e.g. Schwarz et al., 2018; Urraca et al., 2020, 2024). In addition, the SARAH data records have also been used extensively for the analysis of the solar energy resources, its temporal and spatial variability, and the modeling of the energy system (e.g., Huld, 2017; Hörsch et al., 2018; Kaspar et al., 2019; Drücke et al., 2021; Jensen et al, 2023; Sander et al., 2023; Kakoulaki et al., 2024; Husein et al., 2024)

This article provides an overview of the most important aspects of the CM SAF SARAH-3 climate data record. The retrieval algorithm is described in section 2. Section 3 presents the validations of the data record and in section 4 some example applications of the SARAH-3 data record are given. Data availability is described in section 5. Finally, summary and conclusions are presented in section 6.

## 2      SARAH-3 parameters and retrieval method

SARAH-3 is a climate data record generated and distributed by the EUMETSAT Satellite Application Facility on Climate Monitoring (CM SAF). It is the latest edition of SARAH data records and is based on instruments onboard the series of METEOSAT geostationary satellites including the first (MFG) and second (MSG) generations. SARAH-3 thereby combines the MVIRI (on MFG) and the SEVIRI (on MSG) sensors and covers the time period from 1983 to date. The data record covers the region from 65°S to 65°N and from 65°W to 65°E (see Figure 1) and is provided on a regular 0.05° x 0.05° grid. The available temporal resolutions are 30-minutes (instantaneous), daily and monthly means. Figure 2 shows the sunshine duration climatology for Europe. The annual sunshine duration in Europe varies between less than 1000 hours in the North and more than 3000 hours in the Mediterranean area. The parameters included in SARAH-3 are presented in Table 1. Photosynthetic Active Radiation (PAR) and Daylight (DAL) represent specific spectrally-weighted radiation quantities and have not been provided in previous editions of SARAH. PAR corresponds to the part of the solar radiation that can be used by plants to drive photosynthesis; DAL is defined as the brightness [Lux] the human eye is observing. PAR is relevant for biological applications (e.g., oceanic carbon uptake), while DAL can serve infrastructure planning. The spectral weighting used to derive PAR and DAL is presented in Section 2.2.

The concept of SARAH-3 includes the generation and provision of a temporally stable and very consistent climate data record (from 1983 to 2020) based on high-quality and homogeneous input data (i.e. quality-checked satellite data and reanalysis data) and of a near-realtime so-called interim climate data record (ICDR). The ICDR data, starting in 2021, are generated with the same algorithm (whenever possible) and comparable input data as the climate data to ensure a high consistency between the ICDR and the CDR, but provide the data with a timeliness of a few days.

Even though the SARAH-3 CDR can be extended with the ICDR in a consistent way, care should be taken when the CDR-ICDR transition is included in the time series. The main differences between the CDR and ICDR processing are pointed out in the next subsections. More details on the algorithm and validation of the SARAH-3 CDR and ICDR data can be found in the data record documentations available via https://doi.org/10.5676/EUM_SAF_CM/SARAH/V003.

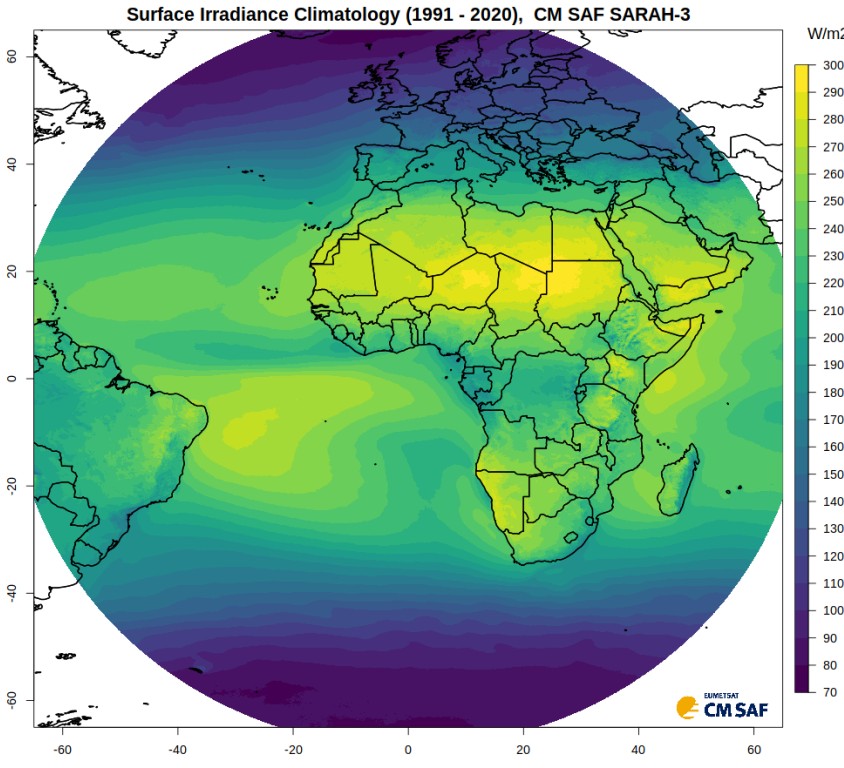

**Figure 1: SARAH-3 surface irradiance climatology for the climate normal period (1991-2020).**

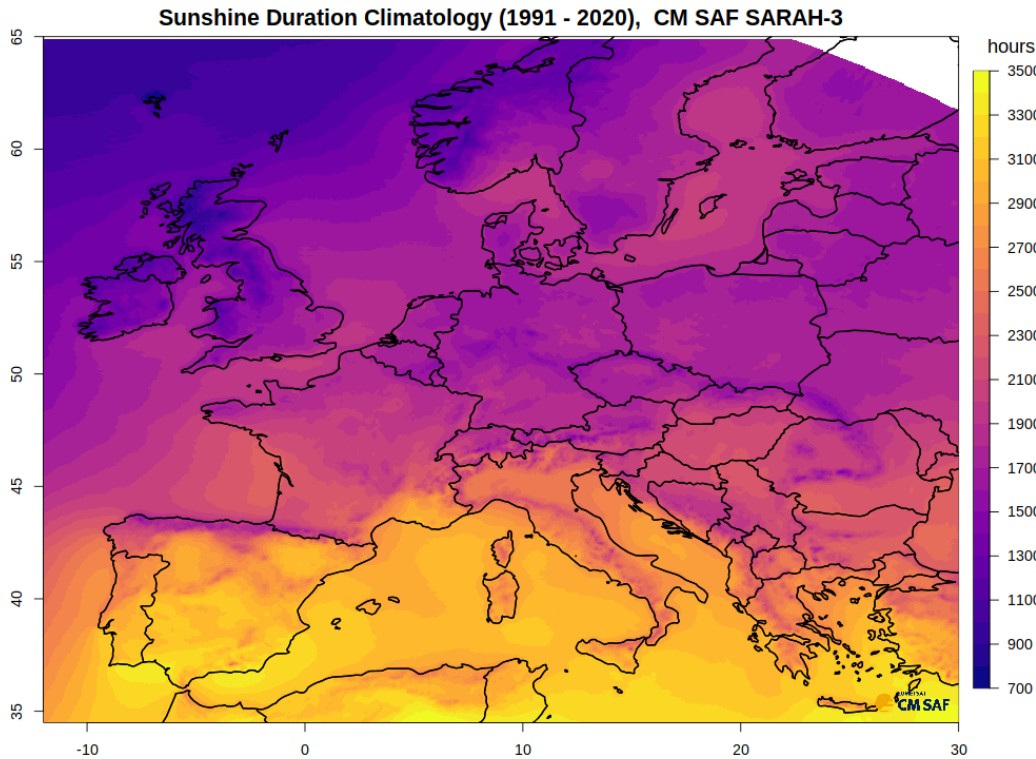

**Figure 2: SARAH-3 sunshine duration mean annual sum for Europe, for the climate normal period (1991-2020).**

| SARAH-3 Parameter | Abbreviation | Unit |
|---|---|---|
| Surface Irradiance (Global Radiation) | SIS | $W/m^2$ |
| Surface Direct Irradiance | SID | $W/m^2$ |
| Direct Normal Irradiance | DNI | $W/m^2$ |
| Photosynthetic Active Radiation | PAR | $\mu mol/(m^2 * s)$ |
| Daylight | DAL | kLux |
| Effective Cloud Albedo | CAL | - |
| Sunshine Duration | SDU | hours |

**Table 1: Parameters included in SARAH-3, with abbreviations and units.**

The retrieval method to estimate surface solar radiation used for the generation of all editions of the SARAH data record is based on the Heliosat-approach (Cano et al., 1986; Hammer et al., 2003) and is described in detail in Müller et. al, 2015b and further put into perspective in Müller et al., 2022. In brief, the method is a two-step approach: First the Effective Cloud Albedo (CAL) is derived from the visible satellite channels only, in a second step CAL is used together with a clear-sky surface solar radiative transfer model to derive the all-sky surface solar radiation parameters. The estimation of the clear-sky surface solar radiation requires some auxiliary data (see Section 2.5). For consistency reasons the visible channel(s) only approach is used throughout the satellite generations, to account for the limited available spectral channels from the MVIRI instrument onboard the first METEOSAT generations satellites.

One main new implementation in the SARAH-3 retrieval scheme compared to previous editions of SARAH is the improved consideration of snow-covered surfaces by internally detecting snow-covered surfaces (see Section 2.1). This information is used as part of the Heliosat-algorithm to generate a more accurate Effective Cloud Albedo in the case of snow-covered surfaces. By combining the SPECMAGIC clear-sky model (see Section 2.2) with CAL, the all-sky surface solar radiation parameters are derived (see Section 2.3). Section 2.4 introduces the sunshine duration parameter and its retrieval algorithm based on the direct normal irradiance (DNI). For the estimation of the clear-sky surface solar radiation using a radiative transfer model some

auxiliary data are required and described in Section 2.5. The estimation of daily and monthly averages from the instantaneous
satellite retrievals is presented in Section 2.6.

## 2.1    Heliosat - HelSnow

Data from the previous editions of the SARAH data records suffered from occasional misclassifications of snow-covered
surfaces as clouds, which resulted in a too high effective cloud albedo (CAL), in particular under predominantly clear-sky and
snow-covered conditions, and subsequently in significant underestimations of surface solar radiation (Niermann et al., 2019;
Carpentieri et al., 2023). With the help of HelSnow, the data quality has improved considerably under such conditions in
SARAH-3 (see Section 3.2).
With SARAH-3, the classical Heliosat approach to generate CAL is extended by the so-called HelSnow-algorithm. The
HelSnow-algorithm is applied to estimate the surface reflectance (rho_min) in the presence of snow before the application of
the 'classical' Heliosat-algorithm. The snow detection in HelSnow is a novel method to efficiently distinguish between clouds
and snow-covered surfaces based on the detection of moving bright objects. This method takes advantage of the high temporal
frequency of observations from geostationary satellites and from the fact that clouds typical move in time, while snow-covered
surfaces are immobile. The HelSnow-method is able to separate snow and cloud coverage based on data from only the
satellite's visible channel, allowing the consistent processing across multiple generations of satellite instruments.
The basic assumption for snow detection in HelSnow is rather simple: Bright areas that are in motion are considered being
cloudy; bright regions without motion may be snow-covered surfaces. As the final result, daily information of snow-covered
surfaces and their daily-averaged brightness is generated, which is used subsequently in the estimation of the effective cloud
albedo. There are four main steps in the implementation of the HelSnow-algorithm to generate daily snow brightness data,
which is subsequently used in the Heliosat-approach.

### 2.1.1    Step 1: Detection of motion

Optical flow is a method from image processing that can detect and quantify motion of objects from a sequence of images.
Using the 'Farnebaeck'-algorithm (Farnebäck, 2003) in standard settings, 'motion' (i.e. the optical flow) is detected in a
sequence of two images (technically the OpenCV software library is used, see https://opencv.org/) in units of pixels per image
sequence. For HelSnow we assume that if the speed of the motion is lower than a certain threshold, the pixel (or objects of
several pixels) is potentially cloud-free. This threshold is different for the MVIRI and SEVIRI sensors (i.e. 0.63 pixel/30 min
and 0.44 pixel/30 min, respectively) due to the different native spatial resolutions of the sensors. An example of the calculated
optical flow speed is shown in Figure 3. All pixels with motion levels above / below the specified threshold are considered in
motion and not in motion, respectively. Only those pixels below the threshold, i.e., those determined to be not in motion, hence
being cloud-free, are further considered.

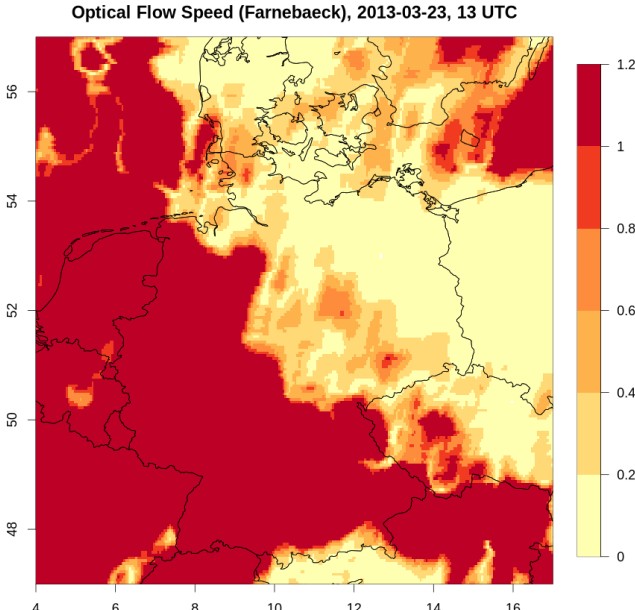

**Figure 3: Optical Flow speed (pixel per 30-minutes) derived by the Farnebaeck algorithm for 2013-03-23, 13 UTC.**

### 2.1.2    Step 2: Detection of sub-daily snow

In the second step of the HelSnow algorithm potentially snow-covered surfaces are identified for every 30-min satellite slot between 0900 UTC and 1530 UTC. For all pixels identified as not-in-motion, i.e., cloud-free, in step 1 the difference between the actual measured reflectivity and a reference clear-sky is calculated. In case this difference is larger (i.e., the pixel is brighter) than a predefined threshold the corresponding satellite pixel is considered snow-covered for this time step / satellite slot, otherwise this pixel is considered snow-free. The reference clear-sky value is calculated for each year based on the individual satellite slots and based on the months of June, July and August. This calculation is done for each year to account for different instrument calibrations and degradations. For the ICDR (2021 onwards) the clear-sky values for 2020 are used, as the SEVIRI instruments are quite stable over time.

In case of clouds (i.e., 'motion' is detected in step 1) the view onto the Earth's surface is not possible, in this case, the last valid observation of the surface for the corresponding satellite slot (e.g. the 1300 UTC slot) (either snow-covered surface or not snow-covered surface) is kept unchanged from the same satellite slot from the previous day. This step is performed for each available satellite measurement between 0900 and 1530 UTC. An example of the instantaneous (snow) reflectivity for 2013-03-23, 13 UTC is shown in Figure 4 (left). The corresponding clear-sky reference value to which the values from the determined clear-sky pixels are compared to, is shown in Figure 4 (right). Note that the reflectivities of snow-covered surfaces typically are substantially larger than those of the reference surface reflectivities. The corresponding threshold used to separate snow-covered surfaces from non-snow-covered surfaces is set to 60 counts.

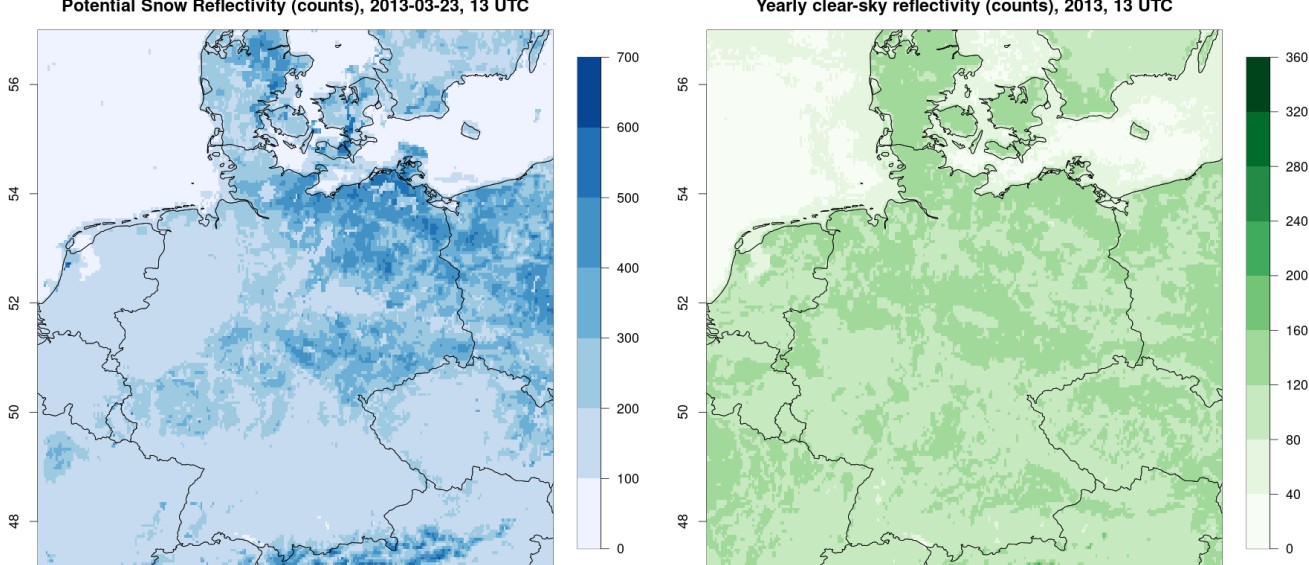

**Figure 4: Example of an instantaneous (potentially snow) reflectivity (left) for 2013-03-23, 13 UTC, and the corresponding clear-sky background reference reflectivity (right) for 2013, 13 UTC (note the different ranges of the color scales).**

### 2.1.3    Step 3: Derivation of the daily snow brightness data

Using the sub-daily (30-min instantaneous) information on potentially snow-covered pixels, pixels are classified as snow-covered for that particular day if the pixels have been classified as snow-covered (in step 2) for more than 2/3 of the used daytime observations. In this case, the associated clear sky reflection ($\rho_{min}$) for these pixels are derived as the temporal average of the instantaneous clear-sky reflections for the particular day and kept constant throughout the day. As a final step, to minimize incorrectly classified snow-covered surfaces (e.g. during fog events), the daily snow-coverage information is corrected using snow and sea ice coverage data from ECMWF global analysis data records (see Section 2.5.1). That means snow-covered surfaces as detected from the satellite observations are not treated as snow-covered if there is no snow in the reanalysis data; in this case the $\rho_{min}$ data as determined by the classical Heliosat approach are used. Figure 5 (left) shows the final daily snow mask / snow reflectivity on 2013-03-13.

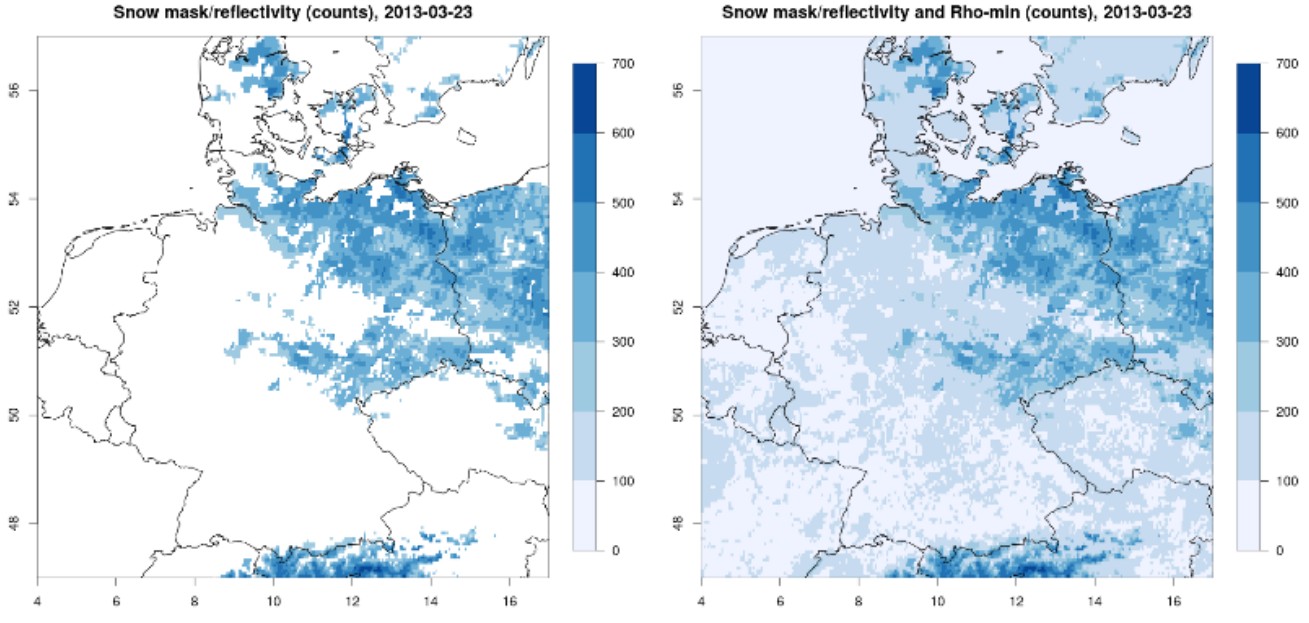

**Figure 5: Daily snow mask/reflectivity (left) and the combined snow mask and rho_min data (2013-03, 13 UTC) (right) used for the derivation of CAL for 2013-03-23.**

#### 2.1.4 Step 4: Heliosat with snow data

The final step of the HelSnow-Heliosat-approach generates the Effective Cloud Albedo (CAL) based on a monthly statistic of satellite images (see also Müller et al., 2015b). The basic formula is $CAL = \frac{\rho - \rho_{min}}{\rho_{max} - \rho_{min}}$. $\rho$ is the actual radiance measured by the sensor, $\rho_{min}$ is the clear-sky reflectance estimated as the minimum reflectance over a certain period of time and derived for each satellite slot to consider the directional surface reflectance. In the case a snow-covered surface was detected by the HelSnow-approach (i.e., allowing the update of the snow-reflectivity in step 3) the daily clear-sky reflectivity is used for all satellite slots. This implies that snow-$\rho_{min}$ is only used under (mostly) clear-sky conditions and prevents the degradation of the sensitivity of the Heliosat-approach under cloudy and snow-covered conditions. $\rho_{max}$ is the maximum reflectance determined per month as derived by the 95[th] percentile of the values in a region in the south Atlantic Ocean with a frequent occurrence of clouds (see also Müller et al., 2015b). $\rho_{max}$ normalizes the cloud albedo and considers the different sensitivities of the satellite instruments and the degradation of the sensor sensitivity in time. Finally, this leads to enhanced temporal stability of the data record.

The result of the HelsSnow-Heliosat-algorithm is CAL, which is the normalized cloud reflectivity relative to the clear-sky reflectance, now considering snow-covered surfaces. CAL is used subsequently as the main input for the calculation of the surface solar radiation parameters.

### 2.2 SPECMAGIC

The SPECMAGIC (Spectral Mesoscale Global Irradiance Code) clear-sky surface solar radiation model is used to estimate the total and direct clear-sky surface irradiance (Müller et al., 2012; 2015b). SPECMAGIC applies an efficient hybrid-eigenvector Look-Up-Table (LUT) approach based on the modified Lambert Beer function (MLB) (Mueller et al., 2004, 2009, 2012) to allow the efficient processing of long-term satellite data. The LUT has been generated using the libRadtran RTM (Mayer et al., 2005). It has been derived for fixed values of integrated ozone, integrated water vapor and surface albedo, two solar zenith angles, and a large range of aerosol properties. SPECMAGIC provides clear-sky surface solar radiation for 32 spectral bands (so-called Kato-bands, see Kato et al., 1999). For more information the reader is referred to Mueller et al., 2012, 2015.

For the calculation of the clear-sky surface solar radiation auxiliary data is required. A description of the auxiliary data used for the generation of SARAH-3 is presented in Section 2.5.

The total and the direct clear-sky surface irradiance are derived as the sum of the irradiances of the 32 spectral Kato-bands. The broadband parameters (SIS, SID, DNI) are calculated by summing up the respective spectral irradiances from all Kato-bands. The clear-sky surface solar radiation for the spectral parameters, PAR and DAL, are derived according to their definitions (see Alados et al., 1995 and https://cie.co.at/) by adding the weighted irradiances from the corresponding spectral Kato-bands. Figure 6 shows the weighting of the Kato-bands for the estimation of PAR and DAL.

### 2.3 All Sky Radiation

The all-sky surface solar radiation is derived by combining the effective cloud albedo derived from the satellite data and the clear-sky surface solar radiation estimated using SPECMAGIC. The clear-sky index, k, is defined as the ratio between the all-sky irradiance I and the clear-sky irradiance I_clr: k = I / I_clr; hence the all-sky surface irradiance is estimated as I = k * I_clr. For the estimation of the surface direct irradiance, the following relation is used: $SID = SID_{clear}(k - 0.38 \cdot (1 - k))^{2.5}$ For more information on the calculation of the direct irradiance we refer to Müller et al., 2015b and Skartveit et al., 1998. The clear-sky index k, can be estimated from the effective cloud albedo using the Heliosat-relation (Hammer et al., 2003); over wide ranges of CAL (-0.05 < CAL < 0.8) the relation between k and CAL is k = 1 – CAL, which provides, multiplied by I_clr,

the estimate of the all-sky surface irradiance: I = (1 – CAL) * I_clr. To estimate the clear-sky index outside this range of CAL
other relations between CAL and k are used (Mueller et al., 2015b).
Spectral effects of clouds are also considered resulting in a spectral adjustment of the clear-sky index, requiring the separate
estimation of the all-sky surface for each individual Kato-band using the spectrally dependent clear-sky index and clear-sky
irradiance. For further information on the estimation of the spectrally-resolved all-sky surface solar radiation parameters see
Müller et al., 2012, 2015b.
The final all-sky irradiance is estimated as the sum of the spectral all-sky irradiances for the corresponding spectral Kato bands,
as described in the previous Section. The Direct Normal Irradiance (DNI) is calculated by $DNI = SID * \cos(SZA)$, where
SZA is the Sun Zenith Angle. The PAR data are provided as Photosynthetic Photon Flux Density (PPFD) in µmol/m2/s, DAL
data are provided in Lux.

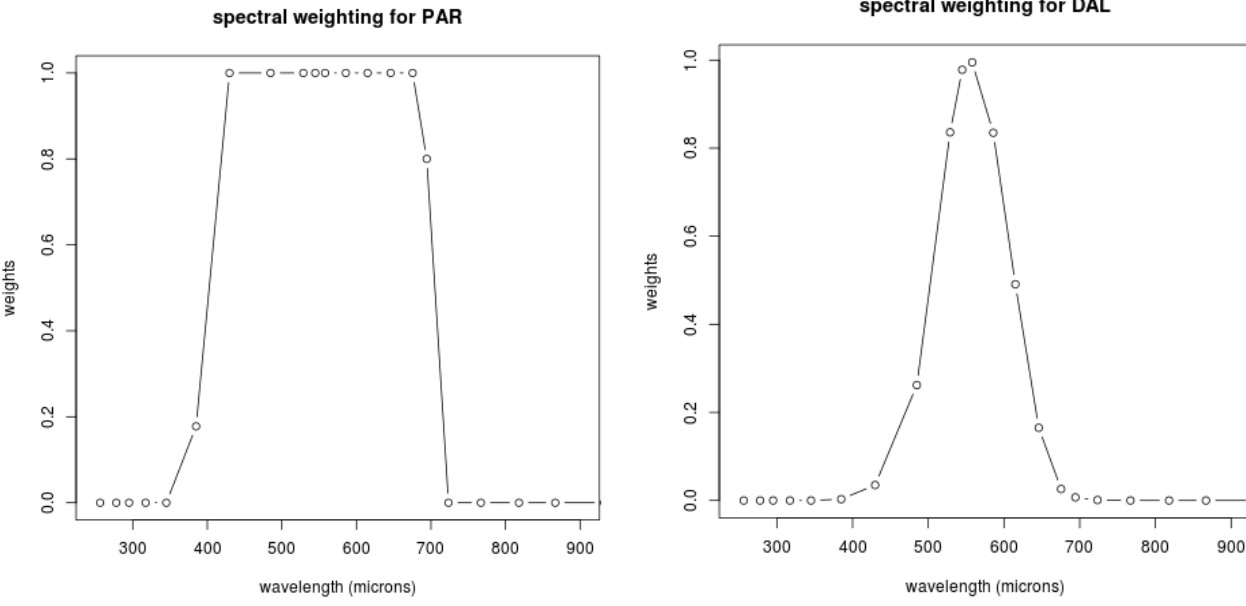


**Figure 6: Spectral weighting for the SARAH-3 parameters Photosynthetic Active Radiation (PAR) and Daylight (DAL).**
**2.4     Sunshine Duration**
Basis for the retrieval of the SARAH-3 SDU data record is the instantaneous (30 minutes) Direct Normal Irradiance (DNI)
data and the WMO threshold for sunshine, which is defined by $DNI \geq 120\,W/m^2$. In SARAH-3 the maximum possible daily
sunshine duration is determined using the 2.5° threshold for the solar elevation angle and the 120 W/m² for the DNI. Here, the
solar elevation angle under clear-sky condition is used and if it falls below the threshold of 2.5°, it is set to exactly the angle
where 120 W/m² is reached. SDU is derived by the ratio of the number of "sunny" satellite slots to all available slots during
daylight multiplied with the theoretically possible daylength:
$$SDU = daylength * \frac{\sum_{i=1}^{iday}(W_i(sunny\_slot_i))}{\#daylight\_slots}$$
The theoretically daylength is pre-calculated depending on the date and location using the simplified SOLIS clear sky radiation
model to estimate clear sky DNI (see Ineichen, 2008, Antonanzas-Torres et al., 2019) and monthly climatological aerosol and
water vapor information. For each day and grid box the length of the period with $DNI_{clr} \geq 120\,W/m^2$ and SZA > 2.5° is
determined and considered as the theoretically possible daylength (Figure 7).
$W_i$ indicates the weighting of sunny slots depending on the number of surrounding cloudy and sunny grid points, which is
discussed in more detail in Kothe et al., 2017, and remained unchanged to SARAH-2.1. The number of daylight slots
(#daylight_slots) describes the maximum number of Meteosat observations (slots) per grid point and per day during daylight
as derived from clear sky estimations of DNI. Daily SDU is calculated only if at least 25 % of the possible daylight slots are
available.

**Clear sky daylength [h] in June**


**Figure 7: Example clear sky daylength [h] based on DNI ≥ 120 W/m² for the 1th of June.**

## 2.5 Auxiliary data

For the generation of the SARAH-3 climate data record a few auxiliary data have been used within HelSnow and for the clear-sky surface solar radiation calculations. Details are covered in the following sections.

### 2.5.1 Snow cover and sea ice thickness

To reduce the number of mis-classified snow-covered surfaces in the HelSnow approach, in particular in the presence of fog, snow-covered surfaces are only considered in the satellite retrieval if snow is present also in global model simulations from ECMWF, which use a wide range of satellite data as well as temperature information from the model simulations to determine the snow-coverage of the surface.

Here, snow cover and sea ice data are combined and used to correct for erroneous daily snow information derived from HelSnow. The global data are remapped to the spatial grid of the SARAH-3 data record. For the CDR time period of the SARAH-3 data record (i.e., 1983-2020) daily 12 UTC data from ERA5-Land (snow coverage) and ERA5 (sea ice cover) (C3S, 2017) are used. Snow and sea ice are considered in case its coverage is higher than 50% for a certain pixel. For the period after 2021 (ICDR processing) the corresponding parameters are taken from the ECMWF IFS operational high-resolution forecast model (IFS model) which deviate from the used ERA5 parameters. For the ICDR, snow depth and sea ice thickness are used if its respective value is at least 5 cm for the grid box mean. This has been shown to deliver mostly equivalent snow and seas ice masks to ERA5. Snow-coverage is only considered in the satellite retrieval if detected by the HelSnow approach; snow information is not added from auxiliary data alone.

### 2.5.2 Water Vapor

The daily Total Column Water Vapor (TCWV) data from ERA5 is used for the CDR. For the ICDR (2021 onwards) the TCWV data is used from the ECMWF IFS operational high-resolution forecast model. Thereby a daily mean is generated from 4 sub-daily fields (i.e. 0, 6, 12 and 18 UTC). As the ERA5 data has a spatial resolution of 0.25° x 0.25°, the TCWV is topographically

downscaled to 0.05° x 0.05° assuming a scale height of ~1600m (see Bento, 2016). For the ICDR processing the TCWV from
the IFS model is used on the native grid with a spatial resolution of 0.1° x 0.1°. Like in the CDR, a daily mean is calculated
and used in the ICDR.

### 2.5.3 Ozone

In SARAH-3 daily mean values of the total vertically-integrated ozone column from ERA5 are used in a spatial resolution of
0.25° x 0.25°. For the ICDR processing, daily mean total ozone from the IFS model with a spatial resolution of 0.1° x 0.1° is
used, similar to the water vapor data, excluding the downscaling step. The data are used in Dobson Units.

### 2.5.4 Aerosols

An aerosol climatology of the European Centre for Medium Range Weather Forecast – MACC (Monitoring Atmospheric
Composition and Climate, see Inness et al., 2013) is used in SARAH-3 (it had also been used for the generation of SARAH-1
and SARAH-2 (see Träger-Chatterjee et al., 2014)). The original MACC climatology has been adjusted to account for the
detection of high aerosol loadings in the HelSnow retrieval based on the study of Müller et al., 2015a and 2015b.

### 2.5.5 Surface Albedo

New data of the surface albedo have been used in SARAH-3 compared to previous editions of SARAH for the estimation of
the clear-sky surface radiation. Here, monthly climatological surface albedo information based on MODIS and prepared by
Blanc et al., 2018, is used. This data is based on Bi-directional reflectance distribution function (BRDF) retrievals given by
MODIS satellite observations. The surface reflectance is provided at a spatial resolution of 0.05° x 0.05° for five spectral
bands. The albedo values from the five spectral bands have been transferred to match the Kato-bands in the SPECMAGIC
clear sky radiative transfer model. This new monthly surface albedo background climatology used in SARAH-3 represents a
substantial improvement compared to previous editions of SARAH, which used surface albedo data based on land-use classes
without monthly variability at a much coarser spatial resolution (0.5°).

### 2.6 Daily and monthly mean generation

The retrieval of the surface solar radiation parameters and the effective cloud albedo is conducted for the whole time period
from 1 January 1983 with a temporal resolution of 30 min; the satellite slots of HH:00 and HH:30 are used for the MVIRI and
SEVIRI instruments, respectively. To ensure the temporal consistency of the data record, no additional satellite slots have been
used from the SEVIRI instrument, which does provide the satellite data with a temporal resolution of 15 min.
The daily means of the surface solar radiation data are based on the 30-minute instantaneous data, using the method by
Diekmann et al., 1988. The formula considers the diurnal cycle of surface solar radiation by using the daily-averaged and the
instantaneous clear-sky radiation:

$$SSR_{DA} = SSR_{CLSDA} \frac{\sum_{i=1}^{n} SSR_i}{\sum_{i=1}^{n} SSR_{CLS_i}}$$

$SSR_{DA}$ is the daily average of SSR. $SSR_{CLSDA}$ is the daily mean clear-sky SSR (derived using SPECMAGIC every 15 minutes),
$SSR_i$ and $SSR_{CLS\,i}$ are the satellite-derived SSR and model-simulated clear-sky SSR for the satellite slot i, respectively. The
criteria for generating a daily mean is that at least 25% of possible daytime pixels must be available (similar to the SDU
generation), otherwise the daily mean data is set to missing for that pixel. The daily averaging is the same for all surface solar
radiation parameters, including the spectral parameters. The advantage of this method to generate the daily means is that the
impact of missing instantaneous data on the daily averaging is much reduced. The effective cloud albedo is arithmetically
averaged to estimate the daily mean.

For the estimation of monthly averages from the daily averages the criteria as defined by WMO for the calculation of monthly means are applied (WMO-No. 1203). These criteria imply that no monthly mean is estimated in case of more than ten daily values or five or more consecutive daily values are missing. If the WMO-criteria are not met, the data will be set to missing for these grid boxes, what occurred for three months for a larger part of the domain (1983-01, 1985-02, 1988-11). The monthly means are calculated by arithmetic averaging of the daily averages.

# 3 Validation

The validation of each data record is an essential mandatory step that each CM SAF data record undergoes before its release. The validation of SARAH-3 is documented in the CM SAF Validation Report available via https://doi.org/10.5676/EUM_SAF_CM/SARAH/V003. Here we summarize the validation of the SARAH-3 CDR and ICDR with surface reference measurements. We further compare the SARAH-3 data record with its predecessor SARAH-2.1, which provides data from January 1983 until May 2023.

## 3.1 Reference data

In this section the reference data used for the validation is described. Surface measurement are used to assess the quality and to validate the SARAH-3 data, as those usually offer the best data quality and can serve as reference.

### 3.1.1 Baseline Surface Radiation Network (BSRN)

The Baseline Surface Radiation Network (BSRN) is a widely used, high-quality network for surface radiation measurements (Driemel et al., 2018, https://bsrn.awi.de/) maintained by the Alfred-Wegener-Institute (Helmholtz-Zentrum für Polar- und Meeresforschung) in Bremerhaven, Germany. The stations are globally distributed, but their overall number is quite small (51 active stations at the end of 2023). The BSRN data include global, direct and direct normal solar radiation data, at most stations with a temporal resolution of 1 minute and are collected with standardized high-quality measurement devices. For the validation of the SARAH data records those 1-minute data are averaged to daily and monthly means using the "M7-method" as recommended by Roesch et al., 2011, that make use of the diurnal cycle of surface radiation to better account for missing values. The BSRN archive provides data since 1994 from, in total, 76 stations, however, with a changing availability of stations over time. BSRN data are used to assess the accuracy of the SARAH-3 data record; for analyzing the temporal stability of a data record their usability is limited due to the comparable short duration of the time series. Table 2 contains the BSRN stations used here for the validation of SARAH-3 (see section 3.3).

| Station | Short name | Latitude [°] | Longitude [°] | Altitude [m] | Temporal coverage |
|---|---|---|---|---|---|
| Lerwick | ler | 60.13 | -1.18 | 84 | 2001-01 to 2017-07 |
| Toravere | tor | 58.25 | 26.46 | 70 | 1999-04 to 2020-12 |
| Lindenberg | lin | 52.21 | 14.12 | 125 | 1994-10 to 2022-08 |
| Cabauw | cab | 51.97 | 4.93 | 0 | 2005-02 to 2024-02 |
| Camborne | cam | 50.22 | -5.32 | 88 | 2001-01 to 2017-07 |
| Palaiseu Cedex | pal | 48.71 | 2.21 | 156 | 2005-10 to 2022-12 |
| Budapest-Lorinc | bud | 47.43 | 19.18 | 139 | 2019-06 to 2023-09 |
| Payerne | pay | 46.82 | 6.94 | 491 | 1993-01 to 2023-12 |
| Carpentras | car | 44.08 | 5.06 | 100 | 1996-09 to 2018-12 |
| Cener | cnr | 42.82 | -1.60 | 471 | 2009-07 to 2024-01 |
| Sede Boquer | sbo | 30.91 | 34.78 | 500 | 2003-01 to 2012-12 |
| Solar Village | sov | 24.91 | 46.41 | 650 | 1998-09 to 2002-12 |
| Tamanrasset | tam | 22.79 | 5.53 | 1385 | 2000-03 to 2024-03 |

| Reunion Island | run | -20.90 | 55.48 | 116 | 2019-06 to 2024-03 |
| Gobabeb | gob | -23.56 | 15.04 | 407 | 2012-05 to 2024-03 |
| Florinopolis | flo | -27.53 | -48.52 | 11 | 1994-07 to 2022-11 |
| De Aar | daa | -30.67 | 24.00 | 1287 | 2000-06 to 2020-01 |

**Table 2: List of BSRN stations used in the validation, including location longitude, latitude, elevation and temporal coverage.**

### 3.1.2 Global Energy Balance Archive (GEBA)

The Global Energy Balance Archive (GEBA) is a collection of global monthly surface irradiance data (Wild et al., 2017; https://geba.ethz.ch/). GEBA includes data from several hundred stations; many of those provide time series for more than 30 years. The quality of the data in the GEBA archive depends on the data provider; no general quality standards for the measurements are required and no general quality control of the data is applied (as it is done as part of BSRN). To ensure the high data quality of the reference data used here, a careful selection of data from stations from the GEBA archive has been made. The criteria of this selection include a high data availability for the study period, a high spatial representativity of the station location, and a temporally homogeneous data record. The latter was determined by applying homogeneity tests using independent gridded data records as reference; these data have also been used to identify outliers in the monthly surface data, which have been removed from the analysis. The final set of 24 stations, which are used for the stability assessment of SARAH-3, are presented in Table 3. All those stations cover the time period 1983 to 2020.

| Station | Latitude [°] | Longitude [°] | Altitude [m] |
| --- | --- | --- | --- |
| Ajaccio | 41.917 | 8.8 | 4 |
| Belsk | 51.833 | 20.783 | 180 |
| Bratislava | 48.167 | 17.1 | 289 |
| Braunschweig | 52.3 | 10.45 | 81 |
| Churanov | 49.067 | 13.617 | 1122 |
| Clermont-Ferrand | 45.783 | 3.167 | 332 |
| Dijon | 47.267 | 5.083 | 222 |
| Graz | 46.983 | 15.45 | 342 |
| Hradec Kralove | 50.25 | 15.85 | 241 |
| Hohenpeissenberg | 47.8 | 11.017 | 990 |
| Karlstad | 59.367 | 13.467 | 46 |
| Kolobrzeg | 54.183 | 15.583 | 16 |
| Kucharovice | 48.883 | 16.083 | 334 |
| Limoges | 45.817 | 1.283 | 282 |
| Marignane | 43.433 | 5.217 | 4 |
| Moscow University | 55.7 | 37.5 | 192 |
| Perpignan | 42.733 | 2.867 | 43 |
| Praha (Prag-Karlov) | 50.067 | 14.433 | 262 |
| Salzburg-Freisal | 47.80 | 13.05 | 420 |
| Strasbourg | 48.55 | 7.633 | 153 |
| Vaexjoe-Kronoberg | 56.933 | 14.733 | 182 |
| Visby - Aerolog. Station | 57.667 | 18.35 | 51 |
| Warszawa | 50.667 | 20.983 | 130 |
| Wuerzburg | 49.767 | 9.967 | 275 |

**Table 3: GEBA stations used for the validation of SARAH-3, including location longitude, latitude and elevation.**
**3.1.3    CLIMAT – monthly sunshine duration data**
CLIMAT is a set of monthly meteorological measurements shared and distributed from Meteorological Services worldwide.
CLIMAT data are collected and distributed by the Deutscher Wetterdienst (DWD) via the DWD Climate Data Center (CDC,
https://opendata.dwd.de/climate_environment/CDC/). CLIMAT includes the sunshine duration as a standard meteorological
parameter, which is used here for the validation of the SARAH-3 SDU data record.
**3.1.4    ECA&D – daily sunshine duration data**
The 'European Climate Assessment and Data' (ECA&D, https://www.ecad.eu/) provides station-based data of several
meteorological parameters at a daily resolution, including sunshine duration, for Europe (Klein Tank et al., 2002; van den
Besselaar et al., 2015). Here we use daily sunshine duration data from the 'pre-defined subset' as provided by ECA&D; non-
blended time series are used, i.e., the data from all individual stations are used and time series have not been merged in case
of station relocation / closure. As for the GEBA archive, the data quality of the data from the ECA&D data depends on the
data provider, no specific quality standards are applied. Also, the instruments to measure the sunshine duration are different
between the available time series, in particular, for those from different data providers.
**3.1.5    German meteorological stations**
The German Meteorological Service (Deutscher Wetterdienst, DWD) provides high quality observational data via its Climate
Data Center (CDC, www.dwd.de/cdc), mainly for Germany. Here we use daily sunshine duration and snow height data from
a large number of stations throughout Germany for specific validation purposes – in particular for evaluating the data quality
of the satellite data in case of snow cover (Section 3.2).
**3.2    Validation of HelSnow**
The newly developed HelSnow-algorithm aims to detect snow-covered surfaces and improves the ability of the algorithm to
distinguish between cloud- and snow-coverage in the visible-channel satellite data. This is especially relevant for clear-sky
situations, when previous editions of the SARAH data record underestimated the surface solar radiation in the case of snow-
covered surfaces.
Figure 8 shows the case for 23 March 2013, when snow cover and clear-sky conditions occurred in Germany and neighboring
regions. The figure shows the improvement of the quality of the sunshine duration data from SARAH-3 compared to SARAH-
2.1 (compare Figure 8, top row). In particular in the north eastern part of Germany, marked by the black circle (where clear-
sky prevails), the SARAH-3 sunshine duration compares much better to the surface reference data than the SARAH-2.1 data.
In this area the snow-covered surfaces were well detected by the HelSnow-algorithm (Figure 8, bottom right). The grey area
(snow detected by HelSnow) agrees to the snow observations from stations (black dots). The data quality improvement is also
shown by the scatter plot (Figure 8, bottom left): The SARAH-3 SDU (red dots) alines much better with the 1-to-1 line than
the SARAH-2.1 SDU (blue dots); the mean absolute differences between SARAH data and the surface measurements drop
from about 2.5 h (SARAH-2.1) to about 1.8 h (SARAH-3).
A similar improvement in the data quality of the SARAH-3 surface irradiance data records is documented in Figure 9 for the
springtime climatological distribution of surface irradiance in the European Alpine region. Figure 9 shows a comparison of
surface irradiance climatologies of March derived from the SARAH-3 and the SARAH-2.1 climate data records compared to
surface reference observations in the European Alpine region extracted from the GEBA. Overall, in the considered regions
SARAH-3 shows higher climatological surface irradiance levels compared to SARAH-2.1, which agrees much better to the
levels derived from the surface reference measurements.

The ability of the HelSnow-algorithm to detect snow-covered surfaces can be determined by comparison with surface observations of snow height / coverage. Here we use data of snow height for Germany from the DWD network, which is available for the temporal coverage of the satellite data record. Figure 10 shows the results of the comparison between the satellite-derived snow mask and the surface measurements for all winter seasons from 1983 to 2019 using the categorical ACC score, defined as the number of correct detections (snow and no-snow) over all cases, and the mean number of days with snow for each season. Overall, the high levels of the ACC-score (median value for almost all years > 0.8) indicate a good quality of the snow mask. A reduced ACC score is correlated with a larger number of days with snow, indicating an underestimation of snow detection by HelSnow. It is worth noting that this evaluation includes situations with snow coverage under cloudy sky; in such situations a snow detection is not possible from the satellite data in the visible channel and the information on snow coverage is estimated from the previous day. The surface solar radiation retrieval, however, is not using the snow information on cloudy days (see section 2.1.4).

The quality of the internal snow-mask slightly improves over time, but is rather stable since the early 1990s. (Figure 10). The reason for the reduced quality of the snow detection in the early years of the SARAH-3 data record is the reduced quality of the satellite input data from the early METEOSAT satellites (less stable, many missing data), which negatively affects the snow detection capability, and the high number of days with snow coverage, which also influences the accuracy of the HelSnow-algorithm. This reduced snow detection quality results in an underestimation of snow and in a more frequent misclassification of snow- as cloud-coverage, which subsequently might lead to a more frequent underestimation of surface solar radiation in the early years of the SARAH-3 data record.

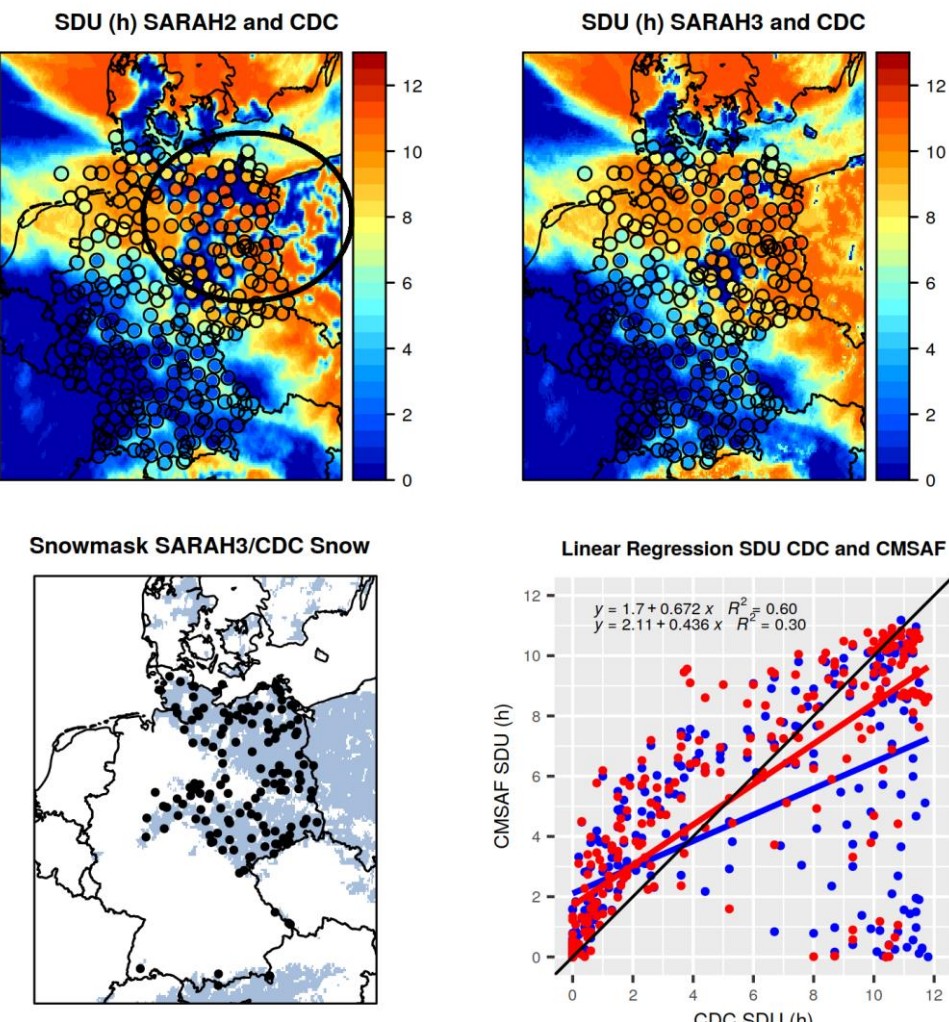

**Figure 8: Comparison of sunshine duration from SARAH-3 (top right) and its predecessor SARAH-2.1 (top left) for a snow case in Germany at 2013-03-23 and comparison to station observations of sunshine duration (dots with same colorbar). The map at bottom right shows the snow cover as detected by HelSnow (grey pixels) and the station data with snow observations (black dots) as overlay.**

**The scatterplot (bottom left) shows SARAH-3 SDU (red dots) and SARAH-2.1 SDU (blue dots) vs the station observations of SDU**
**(bottom right). Included are the linear regressions and the 1:1 line in black. Note that the area of interest in the top left is marked**
**by the black circle, which is the area the snow cover has been misclassified as clouds in SARAH-2.1.**

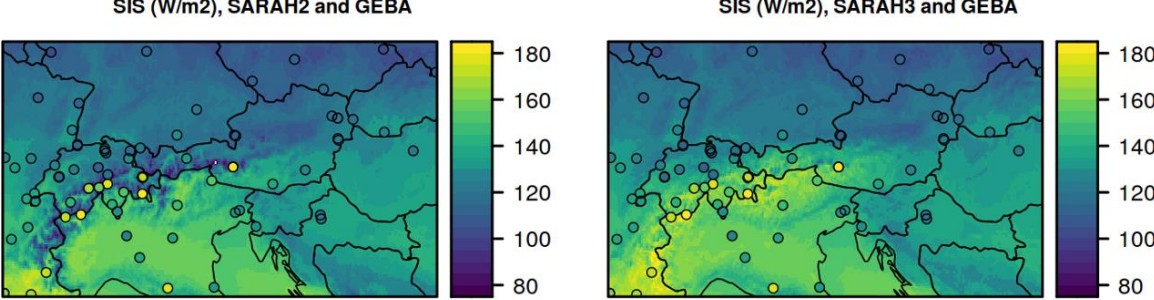


**Figure 9: Validation of surface irradiance (SIS) climatology of SARH-2.1 (left) and SARAH-3 (right) for March together with station**
**observations from GEBA for the alpine region.**

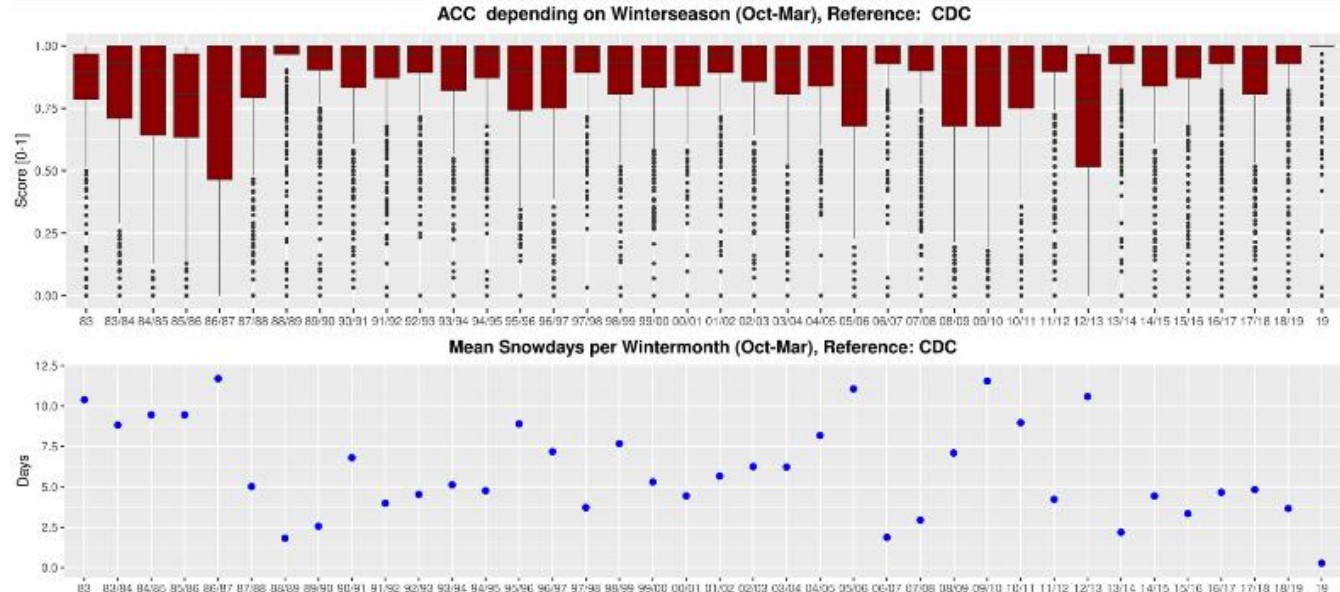


**Figure 10: Time series (1983-2019) of the ACC-Score to validate the snow mask derived by HelSnow with reference to German CDC**
**snow observations. The ACC-score is the measure of the correct (snow or no snow) estimates over all estimates.**
**3.3    Accuracy validation**
**3.3.1    Validation with BSRN data**
Data from the Baseline Surface Radiation Network (BSRN) are the most important reference data source for the validation of
surface radiation data in the CM SAF. The main validation results of the SARAH-3 surface irradiance (SIS) and direct
irradiance data records (SID and DNI) using BSRN data for the time period 1994-2024 are shown in Figure 11 for monthly
and daily averages. For the comparison, data from the SARAH-3 grid box that is closest to the corresponding BSRN station is
used.

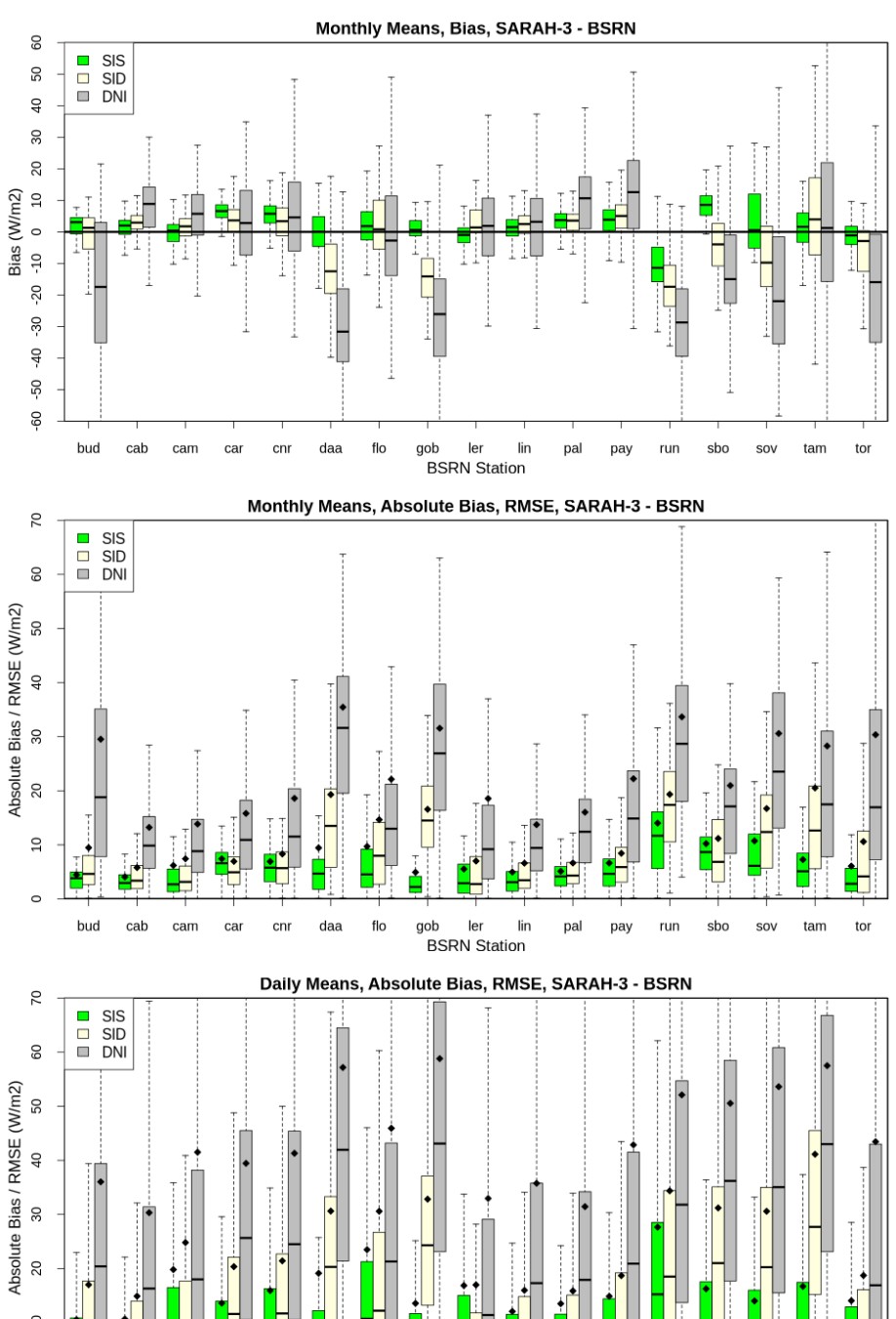


**Figure 11: Validation results of the SARAH-3 parameters surface irradiance (SIS, green), surface direct irradiance (SID, yellow) and direct normal irradiance (DNI, grey) with individual BSRN stations. Shown are boxplots for the monthly mean bias (top), the monthly absolute bias (center) and the daily absolute bias (bottom). The black diamonds plotted onto the box plots in the middle and bottom show the root mean squared errors (RMSE). All data are in W/m². The short names of the BSRN stations are listed in Table 2.**

Figure 11 shows that the bias and the mean absolute differences (MAD) are lower for the surface irradiance (SIS), and higher for the direct irradiance parameters SID and DNI. For the surface irradiance the bias is rather small for most locations, only for the BSRN stations of Reunion Island (negative bias) and Sede Boquer (positive bias) the bias is somehow conspicuous larger than for other locations; the biases are larger, in general, for the direct irradiance parameters (SID and DNI). Concerning the MAD, the situation is comparable (see Figure 11 middle and bottom): The MAD for the direct irradiance parameters are larger than for the surface irradiance. The overall validation results of SARAH-3 vs. BSRN stations for monthly and daily data are summarized in Table 4.


| Parameter | SIS | | SID | | DNI | |
|---|---|---|---|---|---|---|
| temp.res. | mm | dm | mm | dm | mm | dm |
| Bias [W/m$^2$] | 2.1 | 2.0 | 0.5 | 0.5 | -1.6 | -0.2 |
| MAD [W/m$^2$] | 5.2 | 10.8 | 7.9 | 16.1 | 16.9 | 31.1 |
| RMSE [W/m$^2$] | 7.0 | 15.9 | 11.3 | 24.1 | 22.3 | 43.2 |
| Anomaly Cor. | 0.94 | 0.96 | 0.90 | 0.93 | 0.89 | 0.93 |

**Table 4: Summary of validation results of surface irradiance (SIS), surface direct irradiance (SID) and direct normal irradiance**
**(DNI) vs. BSRN stations, for monthly (mm), daily (dm) SARAH-3 data. Shown is the bias, the mean absolute difference (MAD), the**
**root mean squared error (RMSE), and the anomaly correlation (Anomaly Cor.).**
Table 4 shows that the mean bias for all parameters is small with ±2 W/m$^2$. The MAD and root mean squared errors (RMSE)
are lowest for the surface irradiance (SIS) and higher for the direct solar radiation parameters SID and DNI. In general, the
monthly means have lower MAD and RMSE values than the daily means, as daily deviation partly average out over the course
of a month. For the monthly means the MAD for SIS is only about 5 W/m$^2$ and the RMSE is 7 W/m$^2$. The correlations of the
anomalies between the SARAH-3 data records and the BSRN reference data reach and exceed 0.9 for all parameters,
documenting the high quality of the SARAH-3 to identify and to quantify anomalies in the surface solar radiation, which is an
important application for climate data records as well.

### 3.3.2   Validation of Sunshine Duration

Sunshine Duration is a highly relevant climate variable with a long history of surface measurements. It is measured for more
than 150 years and is of high relevance for life. There are many sunshine duration measurements available for validation
purposes. For the SARAH-3 SDU validation we are making use of the monthly CLIMAT data and the daily SDU data from
the ECA&D. The validation results are summarized in Table 5. For the monthly sums, the SDU bias is about 10 hours on
average and about 0.2 hour (or about 12 minutes) for the daily sums. Due to its higher variability concerning day to day
variations, the anomaly correlation of SARAH-3 and the stations is higher for daily sums than for monthly sums. The mean
absolute differences are only about 1 hour for the daily sums of sunshine duration.

| | Bias | MAD | Anom.Cor. | Number Obs. |
|---|---|---|---|---|
| SDU monthly sum | 9.5 h | 20 h | 0.84 | 335.705 |
| SDU daily sum | 0.2 h | 1 h | 0.93 | 10.163.793 |

**Table 5: Summary of the validation of the monthly and daily SARAH-3 SDU with reference to monthly CLIMAT and daily ECA&D**
**sunshine duration data.**
Figure 12 shows maps of the mean bias and mean absolute differences (MAD) per station of the monthly SARAH-3 SDU data
minus the CLIMAT station data. The figure shows that the bias and the MAD are small for most stations, but larger for the
tropical and subtropical stations of Africa. For parts of south eastern Europe, the differences are larger as well. For the majority
of African stations SARAH-3 has a positive bias concerning monthly sums of SDU, reaching values of more than 30 hours.

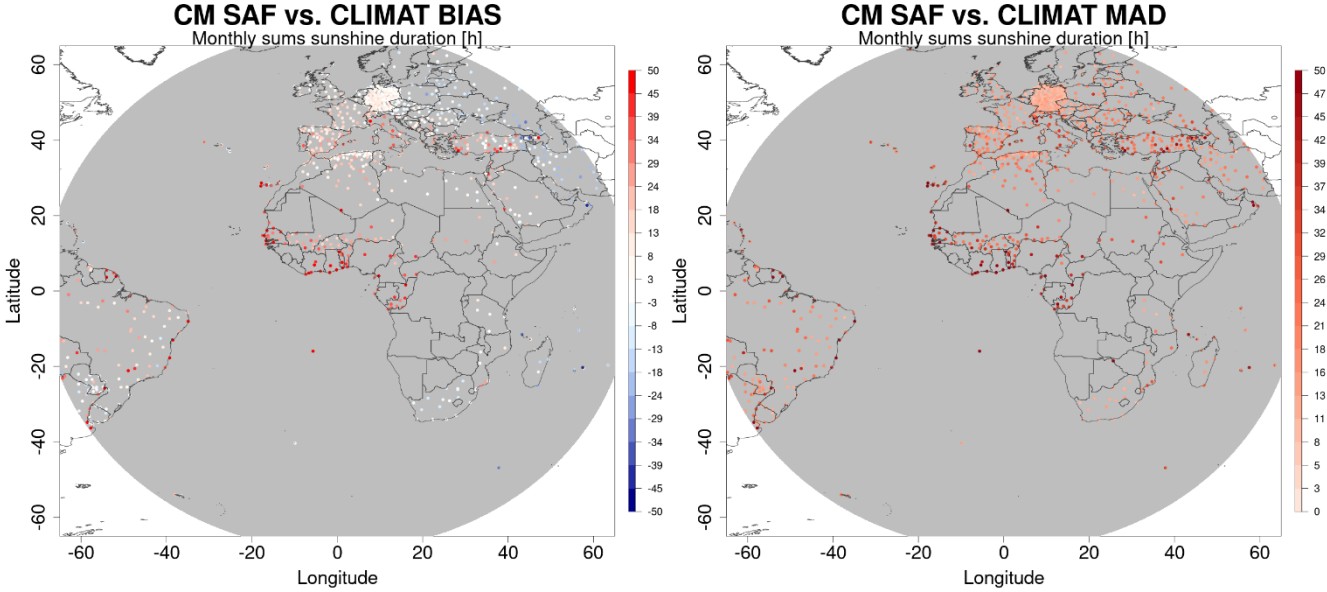

**Figure 12: Map of biases (left) and mean absolute differences (MAD) (right) for monthly sunshine duration from SARAH-3 minus CLIMAT stations.**

### 3.4 Stability validation

### 3.4.1 Sunshine duration validation with CLIMAT

The availability of the long times series of sunshine duration in the CLIMAT data archive allows the analysis of the temporal stability of the SARAH-3 sunshine duration data. The temporal evolution of the bias between the SARAH-3 and the reference data reveals fluctuations and deviations, in particular during the early years of the data record (Figure 13). In the early 1990s there is a period with more positive deviations by SARAH-3, which might be related to the volcanic eruption of Mount Pinatubo on the Philippines in 1991 (Vernier et al., 2011).

The increase of the atmospheric optical depth due to additional aerosols, e.g. by volcanic eruptions, is not directly accounted for in the SARAH retrieval and, hence, might result in an overestimation of SDU in that particular period. The slight and gradual increase of the bias in the mid-2000s is not associated with volcanic activity and requires further analysis. The data quality of SARAH-3 is improved, compared to SARAH-2.1, in terms of the mean bias (~9.5 h vs ~12.3 h) by more than 20% as well as the stability of SARAH-3 as documented by the linear regression lines in Figure 13. Overall there is a slight negative trend in the bias vs the CLIMAT SDU measurements.

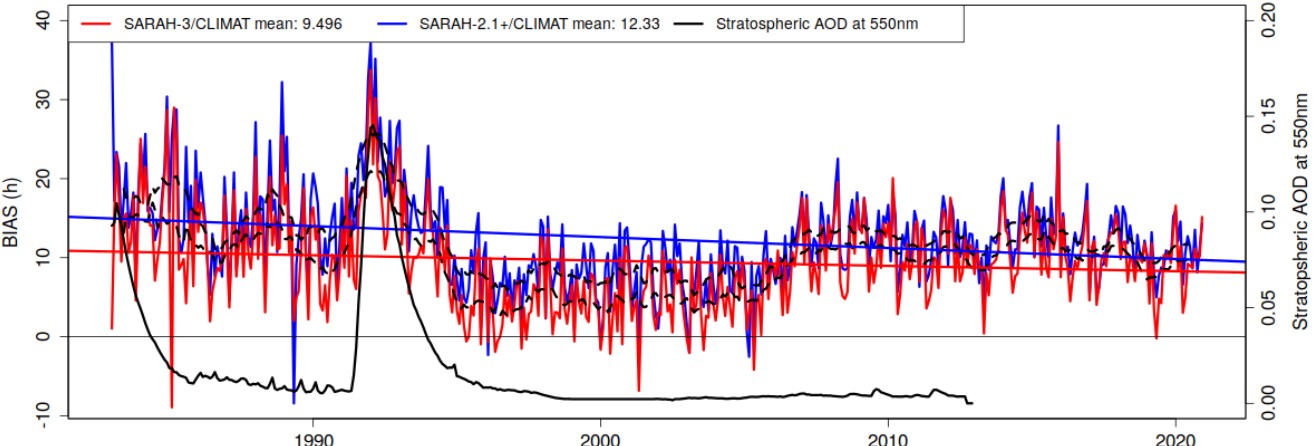

488

**Figure 13: Bias time series of the monthly sunshine duration (SDU) in hours of SARAH-3 vs CLIMAT (red) and SARAH-2.1 vs CLIMAT (blue) for the time period 1983-2020. Additionally, the linear regression lines for both bias time series and the 12-month running means of both bias time series are shown. The black line shows the stratospheric aerosol optical depth (AOD) at 550nm provided by National Aeronautics and Space Administration (NASA) - Goddard Institute for Space Studies (see https://data.giss.nasa.gov/modelforce/strataer/#References for details of the used aerosol data). The mean bias of the SARAH-3 SDU and SARAH-2.1 SDU vs CLIMAT station observations is also provided at the top of the figure.**

### 3.4.2 Surface irradiance validation with GEBA

The monthly surface irradiance data from the GEBA archive is used to assess the long-term stability of the SARAH-3 surface solar radiation climate data record in Europe. Figure 14 (left) shows the time series of the normalized bias between the data from the 24 GEBA stations and the SARAH-3 surface irradiance data record. The numbers at the bottom right of Figure 14 (left) represent the slope of the linear regression line (number in the middle) and its 95% confidence interval (ci), indicating the linear trend of the time series. The 95%-ci defines the range of values, in which the true slope of the linear regression is located with a probability of 95%. The linear trend of the bias based on the 12-monthly running mean time series is -0.64 W/m$^2$/decade, which in turn means that a potential trend in the data from the GEBA stations is underestimated by SARAH-3 by about 0.6 W/m$^2$/decade. The number of stations used for this analysis is rather stable over time due to the used set of selected stations from GEBA (see Section 3.1.2). The number of stations drops to almost zero in February 1985 due to missing data in the SARAH-3 data record for that month as result of the application of the rather strict criteria for the monthly mean generation based on WMO (see Section 2.6).

Figure 14 shows that there is a positive anomaly in the SARAH-3 surface irradiance data record in the early 1990s, which might be related to the Pinatubo volcanic eruption in June 1991. This eruption emitted huge amounts of sulphate into the stratosphere, resulting in the formation of sulphate aerosol, which caused a dimming of the solar radiation in the years afterwards. This dimming by the volcanic aerosols is not accounted for in the SARAH-3 data record, which might cause an overestimation of the surface solar radiation by SARAH-3. A similar behavior in the temporal evolution of the bias has been overserved for the sunshine duration in the early 1990s (see Section 3.4.1). On the other hands, the increase in the surface irradiance bias starts already in 1989, i.e., prior to the Pinatubo eruption, and other factors are likely to contribute to this increase. The decrease of the tropospheric aerosol optical depth due to the reduction of air pollution after 1989 in Europe might also have contributed to the overestimation of surface irradiance by SARAH-3 in this time period.

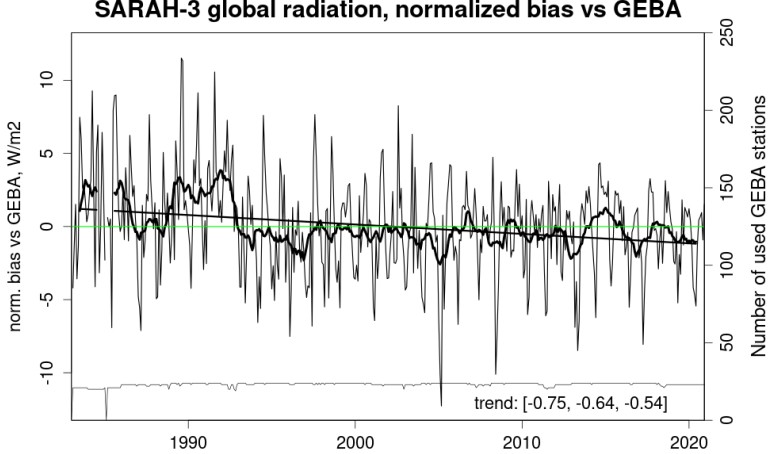

516

**Figure 14: Left side: Time series of the monthly and 12-monthly rolling means of normalized bias (meaning the overall bias of 3 W/m² is subtracted) between the SARAH-3 surface irradiance data record and the GEBA station data for the time period 1983-2020 (black line) including the linear trend line (black) based on the 12-month rolling means. The green line represents the zero-trend line. The grey line (at the lower part) shows the time series of the number of stations used. Additionally, the trend based on the linear regression and its confidence interval are printed (W/m², lower right). Thereby the first number is the lower end of the confidence interval, the 2ⁿᵈ number is the trend and the 3ʳᵈ number is the upper end of the confidence interval. Right side: Map of the GEBA stations used.**

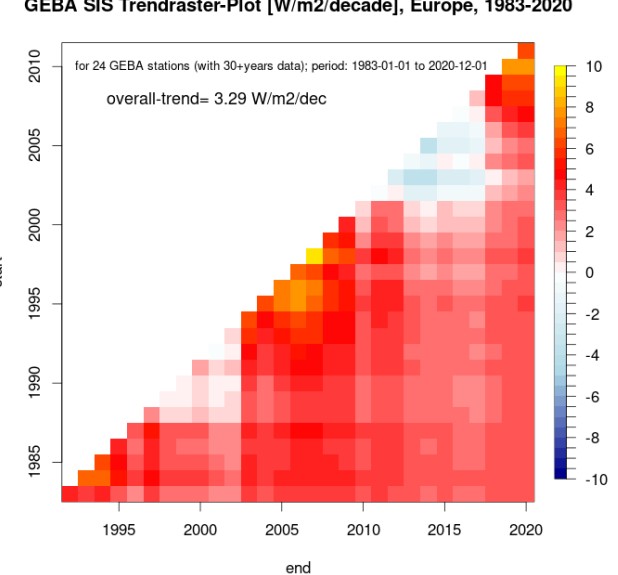

524

**Figure 15: "Trendraster-Plot" of the SARAH-3 (left) and GEBA (right) surface irradiance for the 24 used GEBA stations. Y-axis denote start of trends and x-axis denote end of trends. Trends shown range from 10 years to 38 years (the maximum length of trend, shown in the lower right part of the Trendraster).**

The "running trend" analysis (visualized by so-called "Trendraster"-plots) enables to analyze and to compare variability and trends between two data sets. Figure 15 shows the linear trends over different time period of 10 years and longer; the y-axis denotes the start of a trend estimate and the x-axis denotes the end of a trend estimate. The diagonal shows the shortest (10 year) trends. Figure 15 shows that the temporal pattern of trends as given by SARAH-3 (Figure 15 left) and GEBA (Figure 15 right) are very similar for the average of the used stations. The overall long-term trends of surface irradiance for the period 1983-2020 are also provided in the figure. The trend in SARAH-3 is about +2.7 W/m²/decade and the corresponding trend by GEBA is about +3.3 W/m²/decade. The difference between the trends is about 0.6 W/m²/decade in line with the trend in the bias between both data sets (Figure 14). There is a substantial variability in the decadal trend estimate, which is well represented by the SARAH-3 SIS data record (Figure 15). This variability highlights the high relevance of the start- and the end-year for trend analysis, as can also be seen by patterns (vertical and horizontal lines) caused by the end years 2003 and 2013, that

experience strong positive and negative anomalies of surface irradiance, respectively. In other words, trends ending (starting)
in 2003 tend to be exceptionally positive (negative).

## 540    4       Applications

In this section we will demonstrate some applications of the SARAH-3 climate data record.

### 542    4.1       Climatology

A basic application of a climate data record is the calculation of a climatology by averaging the monthly means for a certain
time period. SARAH-3 covers the current WMO climate normal period from 1991 to 2020; the climatology of surface
irradiance for the full SARAH-3 domain is shown in Figure 1. It shows the typical pattern of maximum surface solar radiation
in the subtropics, in particular in the northern hemisphere and minimum surface solar radiation in the high latitudes. In the
tropics there is a local minimum due to the frequent occurrence of clouds in the Inter Tropical Convergence Zone (ITCZ).
Figure 16 shows the zonal means of all SARAH-3 parameters for the full domain. The meridional variability of the Effective
Cloud Albedo (CAL) is opposite to the surface solar radiation parameters, which follows the relation of CAL to surface solar
radiation, as described in Section 2.3.

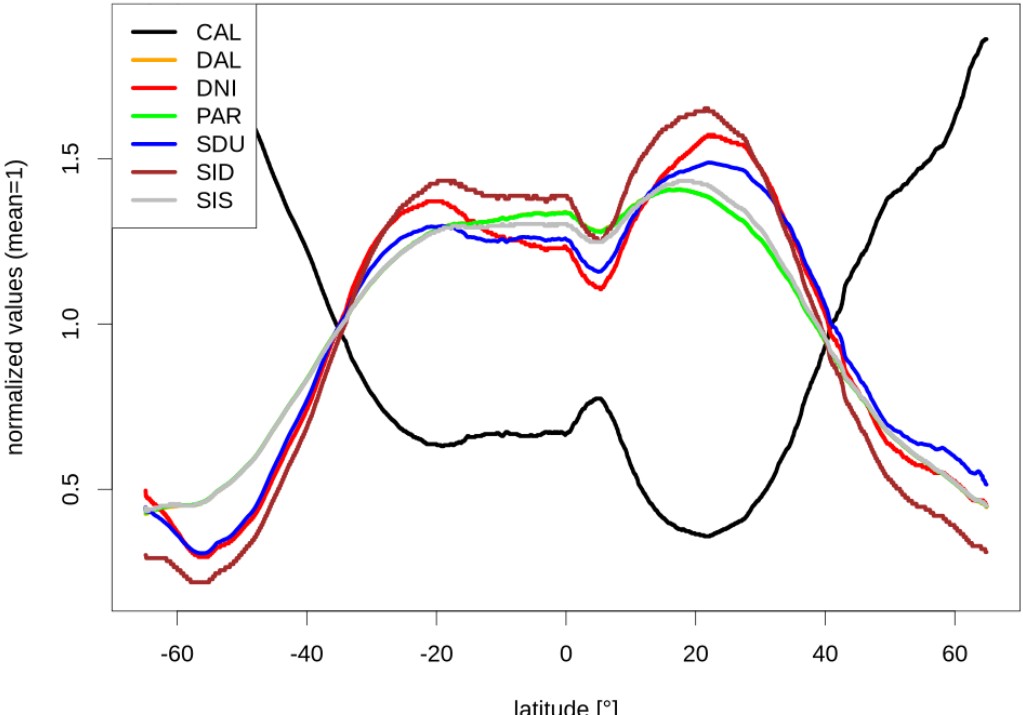


**Figure 16: Zonal means of all SARAH-3 parameters for the full SARAH-3 domain. The parameters are normalized with their respective mean value.**

Figure 16 is meant to give a qualitative overview of the relation of the SARAH-3 parameters by showing zonal means
normalized by its respective climatological mean. All surface solar radiation parameters behave similar concerning the zonal
means, showing maxima in the subtropics and minima in the high latitudes. However, there are also some differences: The
relative surface irradiance (SIS) is larger than the direct radiation (SID) at higher latitudes, where clear-sky situations are less
frequent and hence the contribution of the diffuse radiation is enhanced. On the other hand, the situation is the opposite for the
subtropics, where cloudy days are rare. There, the normalized values for the direct radiation are higher than for the global
radiation (i.e. surface irradiance). A local minimum in all surface solar radiation parameters is visible in the inner tropics (~
5°N), where clouds are relatively frequent due to the convection in the ITCZ. The high cloud coverage south of 40°S results
in low values of the direct radiation parameters (SID, DNI) and the sunshine duration, in particular when compared to the
surface irradiance (SIS), which also includes the diffuse radiation and, subsequently, is impacted less by clouds. The minimum
/ maximum of the effective cloud albedo / the surface solar radiation parameters at about 20°N corresponds to the Sahara
Desert in northern Africa. In general, the anticorrelation of CAL and the surface radiation is obvious. The direct horizontal
radiation (SID) shows the largest meridional gradient.
**4.2 Climate Monitoring**
SARAH-3 is accompanied by an Interim Climate Data Record (ICDR) that consistently extends the Climate Data Record
(CDR) in time. The CDR and ICDR-combination is a powerful tool for climate monitoring applications. The committed
timeliness of the SARAH-3 ICDR is five days, but usually the SARAH-3 ICDR comes with a timeliness of only two days.
Figure 17 shows the spatial distribution of the annual anomaly of sunshine duration for 2022 relative to the climate normal
period (1991-2020). The map shows that 2022 was much sunnier than normal (up to +500 hours of sunshine) in parts of Central
Europe (Germany, BeNeLux, France); parts of the Iberian Peninsula were less sunny than usual in 2022. The SARAH-3
CDR+ICDR combination is used, for example, by the Copernicus European State of the Climate reports (ESOTC; C3S, 2023)
and by the WMO Regional Climate Center (RCC) for the European area (https://rcccm.dwd.de/DWD-
RCCCM/EN/home/home_node.html).

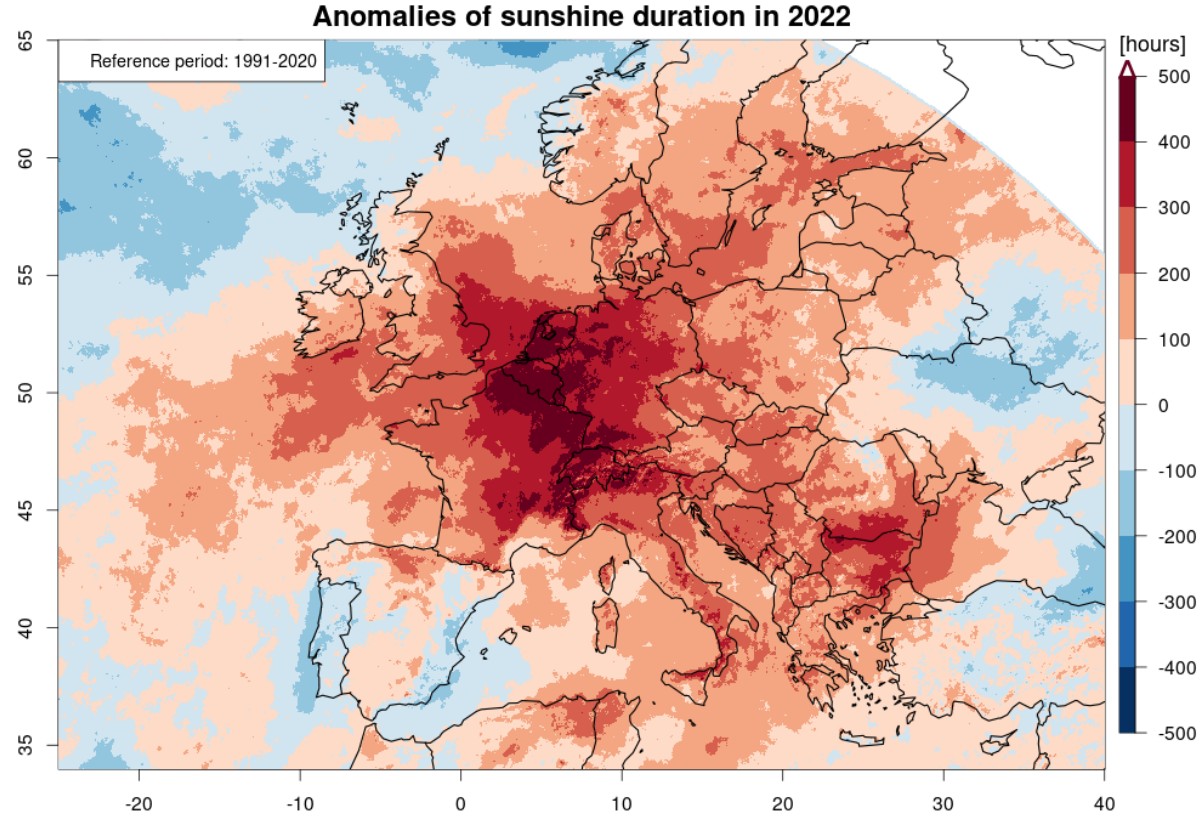


**Figure 17: Anomaly of the SARAH-3 sunshine duration [hours] for 2022, with reference to the climate normal period (1991-2020).**
**4.3 Climate Variability and Trends**
Using a data record for assessing climate variability and trends requires a high level of data quality. Especially the temporal
stability of a data record is crucial for such analyses. Based on the experiences with the previous editions of SARAH (e.g.,
Pfeifroth et al., 2018a) and based on the SARAH-3 validation results, we conclude that it is feasible to calculate trends with a
reasonable confidence, in particular for Europe after about 1990 (see Section 3.4.2). However, it should be mentioned here
that further analyses and validation are required to assess the stability of the SARAH-3 data record for other regions and
periods.
Figure 18 shows the trends of the SARAH-3 surface irradiance (also called global radiation) for the climate normal period
(1991-2020) focusing on Europe. The climate normal period was chosen in order to foster comparability; further, the 1980s
with reduced data quality in satellite and station data are avoided when using the WMO climate normal period. Pixels are only
colored in case the trend is statistically significant. The trend and the significance values are derived using the "trend"-function
from the CM SAF R Toolbox (Kothe et al., 2019). A trend for pixel is considered to be significantly positive (negative) if the
95% confidence interval of the slope of the linear trend (see section 3.4.2 for details) is completely positive (negative).  For
Europe, there are significant positive trends of surface irradiance given by SARAH-3 over the period 1991-2020. Strongest
positive trends are located in Central and Eastern Europe with trends in the range of 2-5 W/m$^2$/decade. Also, parts of the
European Alps stand out with large significantly positive trends of up to 7 W/m$^2$/decade. There, the snow detection by the
HelSnow-algorithm might impact the estimated trend resulting in an overestimation of the trend (see also Section 3.2). There
are almost no significant negative trends of surface irradiance in Europe for the period between 1991 and 2020.

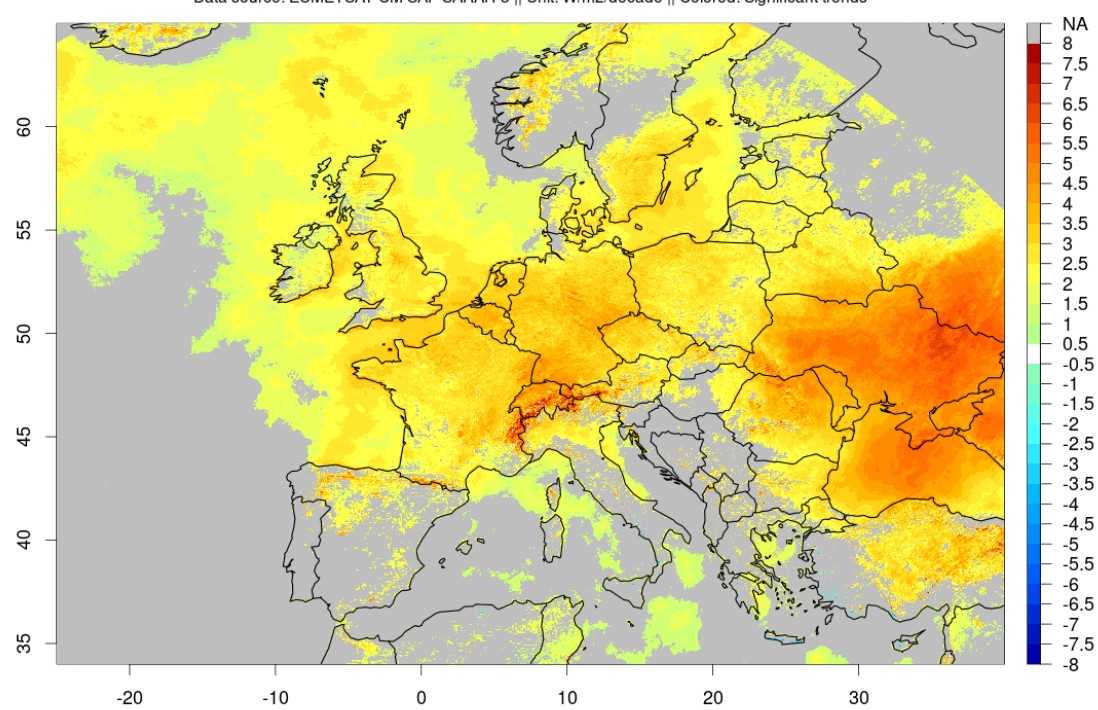

**Figure 18: Trend of the SARAH-3 global radiation in Europe for the climate normal period (1991-2020). Pixels are only colored in**
**case of the trend being statistically significant.**
**5        Data availability**
The data record doi for SARAH-3 is https://doi.org/10.5676/EUM_SAF_CM/SARAH/V003 (Pfeifroth et al., 2023). Data and
associated documentation (scientific references, algorithm theoretical basis documents, validation reports, and user manuals)
are available through the following link: https://doi.org/10.5676/EUM_SAF_CM/SARAH/V003 (Pfeifroth et al., 2023).
All intellectual property rights of the CM SAF SARAH-3 products belong to EUMETSAT. The use of these products is granted
to every interested user, free of charge. If you wish to use these products, EUMETSAT's copyright credit must be shown by
displaying the words "copyright (year) EUMETSAT" on each of the products used.

## 6        Conclusions

SARAH-3 is the new edition of the satellite-based surface solar radiation climate data record (released in May 2023) by the EUMETSAT Satellite Application Facility on Climate Monitoring. SARAH-3 provides data since 1983 (i.e., for more than 40 years) with a spatial resolution of 0.05° and a temporal resolution of up to 30 minutes for Europe, Africa, and parts of Southern America as well as for parts of the Atlantic and the Indian Ocean. SARAH-3 includes seven parameters (see Table 1) including surface irradiance, surface direct radiation parameters, sunshine duration; the Photosynthetic Active Radiation (PAR) and Daylight (DAL) that are new parameters in SARAH-3. The main improvement of SARAH-3 is the improved surface solar radiation estimation in presence of snow cover, which is internally derived by the HelSnow algorithm. Further, several auxiliary data are updated, incl. the surface albedo, which now has a spatial resolution comparable to the SARAH-3 data itself. The SARAH-3 data record and all other data records released by the CM SAF are available free of charge via the CM SAF Web User Interface ([www.wui.cmsaf.eu](www.wui.cmsaf.eu)) in NetCDF-format.

The algorithm used to generate SARAH-3 has been subject to continuous developments since the 1st release of a METEOSAT-based surface radiation data record by the CM SAF, while the basic algorithmic approach (i.e., a Heliosat-based retrieval) has been unchanged. The improved auxiliary data has also contributed to improved final data products, e.g. through the usage of daily ERA5 atmospheric background fields, instead of monthly ERA-Interim data. The new snow detection by HelSnow leads to improved accuracy and reduced biases, especially in case of snow cover and clear-sky conditions (see Section 0).

The validation (see Section 3) shows that SARAH-3 offers high quality climate data; the uncertainty of the data increases with increasing temporal resolutions. The validation of the SARAH-3 direct solar radiation parameters shows higher differences to surface reference measurements than for the surface irradiance (called global radiation). For the latter, the mean absolute differences between the SARAH-3 data and surface reference measurements are about 5 W/m$^2$ and 11 W/m$^2$ for monthly and daily averages, respectively. Note that these measures include the uncertainties of the surface measurements and are impacted by the difficulty of comparing point measurements to grid-box averages. An important validation measure for climate data records is also its ability to detect and quantify anomalies, which is measured by the anomaly correlation. For SARAH-3 the corresponding correlation coefficients are between 0.84 and 0.98, documenting the ability to use the SARAH-3 data for climate monitoring applications (see Section 4.2).

The stability of SARAH-3 has been found to be high and further improved relative to its predecessor. The comparison with long-term surface reference measurements in Europe from GEBA revealed that there is a small negative trend in the time series of the bias between SARAH-3 and surface reference data of about -0.6 W/m$^2$/decade for surface irradiance for the period 1983-2020. Further, trends in the European Alps are likely overestimated by SARAH-3 when considering the full time series of the data record (1983 onwards). The reason for this trend overestimation is the reduced quality of the snow detection by HelSnow for the early years of the data record. For the climate normal period of 1991-2020, and onwards, this issue is strongly reduced, and hence the stability in the Alpine region is improved from the 1990s onwards. The 1991 Pinatubo volcanic eruption likely led to an overestimation of the surface solar radiation and sunshine duration during that period of enhanced aerosol loadings in the stratosphere.

In Section 4 some example applications of the SARAH-3 data record are shown. The climatology of a certain parameter gives insights to the spatial distribution of the respective parameter, which is useful for many applications. For the first time the current SARAH climate data record covers the current climate normal period from 1991 to 2020. In addition, the availability of instantaneous (30-minutes), daily and monthly data and of data from the ICDR, which operationally extends the data record, allows a wide range of applications of the SARAH-3 climate data record, including climate monitoring, see Figure 17, and climate analyses. The interpretation, however, of long-term trends should be done with care, since such trends are strongly influenced by anomalies at the beginning and end of the time series considered. The validation results of SARAH-3 show that the data can be used for trend analysis with reasonable confidence. The linear trend of the SARAH-3 global radiation for 1991-2020 in Europe is overall positive, which is in line with surface observations (see Figure 15).

Future developments of the CM SAF SARAH data record include the transition from the METEOSAT only setup towards the inclusion of other geostationary satellite orbits to provide the data at an improved spatial coverage. The combination of such a data record, i.e. SARAH-GEO, with a data record based on polar-orbiting satellites, e.g., the CM SAF CLARA data record, allows the generation of a multi-satellite multi-platform global data record.

With its numerous surface solar radiation parameters, high quality, long time series, high spatial and temporal resolution and high timeliness (~2 days), the freely available SARAH-3 data record continues to serve users in many fields of research and operation. In case of questions or inquiries regarding the SARAH-3 data (or any other CM SAF data), the CM SAF User Help Desk is available via contact.cmsaf@dwd.de.

**Author contribution**

UP prepared the original manuscript with substantial contributions from JT. JD contributed to the data validation of sunshine duration. UP and JT developed and validated the surface radiation products. UP generated the data record, supported by SK and supervised by JT. MS and RH provided valuable comments and recommendations for the structure of the manuscript. All authors contributed to the manuscript. All authors contributed the writing or reviewing and editing.

**Acknowledgement**

The authors acknowledge the financial support of the EUMETSAT member states through the Satellite Application Facility on Climate Monitoring. Further, the authors like to thank Ruben Urraca for the cooperation on the quality control and usage of the GEBA surface reference measurements. Further we thank the World Radiation Monitoring Center – Baseline Surface Radiation Network (BSRN) and the Global Energy Balance Archive (GEBA) for providing surface reference data. GEBA is co-funded by the Federal Office of Meteorology and Climatology MeteoSwiss within the framework of GCOS Switzerland. We also thank Prof. Elmar Schömer and Kai Wirtz from the Johannes Gutenberg-University of Mainz for the HelSnow development.

**Competing interests**

The contact author has declared that none of the authors has any competing interests.

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
