# Peer review of "SARAH-3 – satellite-based climate data records of surface solar radiation"

_Earth System Science Data, 2024_

## Author Comment (AC1)

ESSD https://doi.org/10.5194/essd-2024-91

**RC1**: 'Comment on essd-2024-91', Anonymous Referee #1, 10 May 2024  reply

The described surface solar radiation climate data records (CDR) product SARAH is important and helpful for understanding the climate system, model evaluation, renewable energy application, etc. This preprint introduces a new version, SARAH-3, which updates the  CDRs and motivates and explains new updates in the applied methods and their impact. The preprint is very well written and illustrated, but I have some questions and suggest some minor improvements to be made.

- Abstract: the reader would like to see the covered region here already (SARAH is not global)

A:  Thanks for the comment. Yes, it is important to mention early on that SARAH-3 is not global. This information will be added to the abstract.

- Abstract: Name the seven parameters of Climate Data Records explicitly here for clarity, as you mention Interim CDRS later in the abstract.

A: Yes, we will name all seven parameters already in the abstract.

- Introduction: for some applications, it might be helpful to use a fitting top-of-atmosphere CDRs dataset. Can the authors suggest a data set?

A: Thanks for this question. There is the global CM SAF CLARA-A3 (DOI) data record available which includes top-of-atmosphere fluxes but is given on different spatio-temporal resolution. There is also the CMSAF "Top of Atmosphere Radiation MVIRI/SEVIRI Data Record" (https://doi.org/10.5676/EUM_SAF_CM/TOA_MET/V001) data record providing daily and monthly means for Feb 1983 to April 2015. This data is one the same spatial resolution covering a slightly larger area. A publication analyzing CM SAF surface and top-of-atmosphere solar radiation is given in the references (Pfeifroth et al., 2018.)

- Line 78: -> "SARAH-3 paramters, abbr & units". It sounds strange that "units are included" in the dataset.

A: Will be revised ro read: "Parameters included in SARAH-3, incl. their abbreviations and units

- Sec. 2.1: Is there no reference available for HelSnow?

A: Unfortunately, there is no dedicated reference for HelSnow available. HelSnow is first introduced and explained in this publication.

- Line 114: Why no units? What is 160? I guess pixels, but guessing is risky. So, is the displacement speed more than 160 pixels/30 min in the case of MVIRI?

A: You are right, Thank You for this comment! – we forgot to add some essential explanations concerning the numbers. Indeed, the unit of the optical flow speed generally is pixels per image sequence (30min in this case). For our application we wanted to have

much more contrast at low speeds, so the resulting optical flow speed in truncated at 1 and stored as 8-bit greyscale image with a maximum value of 255. This means a value 255 corresponds to an optical flow speed of 1 pixel/30 min, and a threshold value of 160 for MVIRI corresponds to a speed of 160/255 ~ 0.63 pixel/30 min. As for SEVIRI the native pixel size of the visible channels is larger, the speed threshold is reduced accordingly to 112/255 ~ 0.44 pixel/30 min. In other words, the threshold used for the optical flow speed is quite low in order to assure clouds are excluded from the snow detection process. These explanations will be added to the updated manuscript.

- Figure 3: units? What is optical flow (a term from image processing?)? Displacement?

A: Thanks for this comment. "Optical Flow" indeed is a term from image processing. It is method that can detect a change of objects from one image to another. One output of "Optical Flow" is the speed of an object or pattern from one time step to the next in units of pixels per image sequence. Different "Optical Flow" Methods are related functionalities are included in the OpenCV software library, which is used for HelSnow. We will add some general information to the manuscript.

- Sec. 2.1.: Is snow ageing and thus the change of snow albedo of relevance? Even considered?

A: Thanks for this comment. You are touching an important point. Actually, it is of special importance not only if there is snow but also how the snow albedo is like. As HelSnow tries to detect snow every day, snow ageing can be detected (in case of clear-sky) and is then considered in HelSnow. However, the snow albedo is kept constant in case no surface (snow) detection is possible.

- Line 146: How much would the results degrade if ERA5 snow-cover were used (after interpolation)? In other words, what is the quantitative added value of HelSnow?

A: The snow mask of HelSnow is of lower accuracy compared to the snow coverage given by ERA-5, which is partly due to the fact the snow can only observed during clear-sky situations. The advantage of using HelSnow is not only to have a snow mask but to also have the actually observed snow reflectivity for each pixel which and time. For example, snow in the forest appears much darker than snow on grassland. The actual snow reflectivity is important to estimate a reasonable effective cloud albedo (CAL).

- Line 187: SID = ??? How derived?

A: For the calculation of SID we would like to refer to Müller et al., 2015 and Skartveit, A., Olseth, J.A. and Tuft, M.A. (1998) An Hourly Diffuse Fraction Model with Correction for Variability and Surface Albedo. Solar Energy, 63, 173-183. We will point to these references for SID in the revised manuscript.

- Do you use ERA5-Land snow cover? The ERA5_Land snow cover does not assimilate snow observations! ERA5 does, but not in complex terrain.

A: Thank you for the valuable comment. Yes, we are using ERA5-Land snow cover for the SARAH-3 CDR. The high spatial resolution and reasonable quality were suited for our purposes.

- Sec. 2.5.4: Is a change in aerosol concentration over time considered? MACC does not

cover the entire SARAH period? How can you discuss trends without including AOD change? A reference for MACC?

A: A reference to MACC will be added to the manuscript. Yes, in SARAH-3 we are using a monthly climatology of aerosol information. This means the direct aerosol effect is constant over time. The indirect aerosol effects (brighter clouds longer lifetime of clouds in case of more aerosols) are included through the clouds itself. Assuming the SARAH-3 data record is homogeneous over time, the surface radiation trend would be underestimated assuming there is a negative trend in aerosol concentration. Overall, we see that for Europe the trends in surface irradiance between SARAH-3 and surface reference measurements fairly agree (see Figures 14 and 15) which indicates that the direct aerosol effect plays a minor role for the observed surface irradiance trends. The majority of trends and decadal variability seems to be cause by changes of clouds. The observed underestimation on the trend in SARAH-3 of 0.6 W/m2 trend might be due to missed direct aerosol effect.

- Table 3: Strassburg -> Strasbourg?

A: Thanks for the comment. Will be changed accordingly in the updated manuscript.

- Line 379: "and its functions"?

A: Thanks. Will be corrected in the updated manuscript.

- Figure 11: Absolute Bias is called MAD elsewhere?

A: Thanks for the comment. We try to be consistent throughout the manuscript. "Absolute Bias" and "MAD" (Mean Absolute Difference) are synonyms. We will go through the manuscript to stick to a consistent wording in the revised version.

- The references list needs to be sorted and, therefore, difficult to check.

A: Thank You for the comment. We will re-order the reference list to be in alphabetic order.

---

## Author Comment (AC2)

ESSD https://doi.org/10.5194/essd-2024-91

**RC3**: 'Comment on essd-2024-91', Anonymous Referee #3, 10 June 2024  reply

This article presents a concise and well-written overview of the theoretical basis and updates of the latest release of the SARAH climate data record. The article is a very useful source of reference information for potential users of this dataset and falls well within the scope of ESSD. There are a few minor points (in addition to/partly overlapping with the points mentioned by the other reviewers), which I ask the authors to address prior to publication.

A: Thank You very much for your constructive review and the positive feedback.

List of minor points:

* Abstract: language- and information-wise, I think that the abstract can still be improved. Specifically, I suggest to mention all parameters included in SARAH-3 (or at least the newly added ones?) and to explicitly mention the start data of the dataset. I do also think that the term SARAH-3 is used to frequently. The mention of „ICDR / near-real time processing" directly before the statement „enabling climate monitoring applications" seems mis-leading, as the near-real time extension of the data record is not really crucial for climate applications. The phrase „The SARAH.3 climate analysis reveals" seems la strange start of this senetence (is this a „standardized" analysis?). „good accuracy and stability": can you give quantitative numbers here, e.g. for SSR?

A: Thank You for these constructive comments concerning the abstract. We will try to improve the writing, by giving some more fundamentals of the data record (parameters, start of data record, etc.). In our understanding a typical climate monitoring application is to calculate the anomaly of last month/year, and this would not be possible without having the continuous consistent extension of the time series. That's way "climate monitoring" is mentioned close to the availability of the ICDR. We will also add some quantitative numbers.

* ICDR: I think the distinction between the CDR and ICDR needs to be described in more detail, including giving some guidance for users of this data record. While the start of the ICDR period is mentioned twice, this is still somewhat hidden in the description of meta-data, and I recommend to dedicate a paragraph to this at the start of Section 2. What is the plan/time-line for updating periods now covered by the ICDR to CDR-quality processing? If one is interested in climate trends, when should one wait for the availability of the CDR instead of calculating trends based on the spliced CDR/ICDR?

A: We agree that the description of the ICDR has been relatively short, and we will add some more information on it in the revised manuscript. There will be no update of the ICDR to CDR. The important thing is, that the ICDR-quality does not fall behind the CDR-quality. The ICDR extends the CDR as long as a new CDR will be available, which will likely will be extended by an ICDR as well. As far as we can see, trends based on CDR+ICDR data are valid, but a dedicated validation is recommended when the transition from CDR to ICDR is included in the time series. More information on the data validation and quality of the SARAH-3 CDR and ICDR can also be found in the SARAH-3 Validation Report via www.cmsaf.eu.

* L55: „All CM SAF data records are freely available without restrictions." While I am not a lawyer, I do not think this statement is true. Specifically, the clause to acknowledge EUMETSAT in the CMSAF license is a restriction as far as I can see (even if a very weak one), or not?! Please check/correct this aspect. (see https://cds.climate.copernicus.eu/api/v2/terms/static/eumetsat-cm-saf.pdf )

A: Thank You for pointing us to this aspect. We shall add the information about the need to acknowledge EUMETSAT CM SAF when used. The phrase "without restricitons" will be removed in the revised manuscript. The correct phrasing from the CM SAF website cmsaf.eu is "All intellectual property rights of the CM SAF products belong to EUMETSAT. The use of these products is granted to every interested user, free of charge. If you wish to use these products, EUMETSAT's copyright credit must be shown by displaying the words "copyright (year) EUMETSAT" on each of the products used." Detailed information on the EUMETSAT license can be found here: https://www.eumetsat.int/data-policy/eumetsat-data-policy.pdf#download=1

* As mentioned in the introduction, homogeneity is a key goal for CDRs. Do you see any impact of transitioning to MSG? Can you also discuss aspects where sensor limitiation affect the algorithm choices. I guess you could have used the 1.6um channel for snow detection from SEVIRI. Does the difference in spatial resolutions between Metesat first and second generation have an impact?

A: Yes, homogeneity is very important for a CDR. We have not seen any major inhomogeneities in the MFG-MSG transition. However, a recent not yet published study by Ruben Urracca just found that there is a small inhomogeneity on the seasonal scale likely related to the MFG-MSG transition, which diminishes on the annual scale. Indeed, sticking to the visible-channel(s)-only-approach is also done for the sake of consistency and homogeneity, as there have been only very limited channels onboard the first METEOSAT generations satellite instrument MVIRI. The different spatial resolutions do not seem to have an impact on the temporal stability.

* I found it surprising that optical flow is used to identify snow in HelSnow, by an exclusion logic (i.e. if it moves, it cannot be snow). Did you consider using a constraint on temporal constancy instead (if reflectivity does not change significantly in time, it must be snow)? In addition, what window size and other parameter settings are used for inferring the optical flow?

A: Thank You for this comment. We did not use the temporal constancy to identify snow. This might be an option for future development, but due to the different illuminations (sun – satellite constellations) ich might be difficult to make use of this criteria. Further also the snow coverage might change of time, especially in case of a thin snow cover or in forest or urban areas, where snow falls down the leaves or is actively removed. For the optical flow we used the Farnebeack algorithm with standard settings through the OpenCV.org software library:  "calcOpticalFlowFarneback(prevgray, gray, flow, 0.5, 3, 15, 3, 5, 1.2, 0);", See https://docs.opencv.org/4.x/dc/d6b/group__video__track.html#ga5d10ebbd59fe09c5f650289ec0ece5af

* While I know PAR, I did not know the term DAL before. Can you please add some additional explanations what these parameters are used for/what spectral weighting is used?

A: DAL (Daylight) is a parameter that comes in [Lux] and is defined as the brightness the human eye is observing. It is a quantity that can serve the infrastructure planning user group like architects. Figure 6 (right) shows the weighting of the spectral bands to derive DAL. Daylight is defined and described here: https://cie.co.at/publications/daylight; Spectral information on Daylight can be found here: http://www.cvrl.org/

* Figure 9: I suggest to replace the SARAH2 image by a difference image (or add this as 3rd panel). I found it hard to quickly identify differences and think a difference image would help to highlight regions with significant changes.

A: Thanks for the suggestion. However, we think that the absolute values in Figure 9 in SARAH-3 and SARAH-2 reasonably show, together with the station data (dots), the better correspondence of SARAH-3 and the stations. We think that it makes sense to keep the 3rd (bottom left) plot as it easily shows the improvement of SARAH-3 over SARAH-2. Figure 9 bottom right shows that the derived snow mask nicely corresponds to the station observations. For better identification of the region of interest we propose to highlight that region by a circle in the updated manuscript.

* Larger context and outlook: I missed a reference to the comprehensive review of Huang et al., 2019 (*). For readers, it could provide additonal context, in particular with respect to your method and their section on „Current problems". Can you add some thoughts on future improvements of SARAH? How will SARAH-3 be affected by the transition to MTG? (*) https://doi.org/10.1016/j.rse.2019.111371

A: Thank You for this values comment. We will refer to the comprehensive paper by Huang et al., 2019. We will also add some more information about future improvements of SARAH and the transition to MTG. In short, we plan to extend the spatial coverage by using also other geostationary satellites than METEOSAT prime. A main purpose of the CM SAF is the generation of long-term, homogenous data records and therefore we will continue to rely on the MVIRI-channel heritage. Developments are ongoing to assure a smooth transition to MTG. As Huang et al., 2019 also points out, a combined multi-satellite multi-platform global data record is a long-term goal.

---

## Author Comment (AC3)

ESSD https://doi.org/10.5194/essd-2024-91

**RC1**: 'Comment on essd-2024-91', S. Kato, 1 June 2024  reply

The authors describe the surface solar radiation data product SARAH-3, which is a revised version of SARAH-2. The product includes global surface irradiance, direct irradiance, direct normal irradiance, photosynthetic active radiation, daylight, effective cloud albedo, and sunshine duration. Computations of these variables are based on parameterized model. Resulting global irradiances are evaluated against BSRN and GEBA data set. The manuscript is a thorough description of the data set. I only have minor comments/suggestions on this version.

A: Dear S. Kato, Thank You for taking note of our manuscript and for Your review and positive feedback.

Line 114: Could you include units of 160 and 112, or convert this to m s-1 if you know the size of the pixel.

A: Indeed, we have been unclear about the optical flow speed. We missed to add some essential explanations concerning the numbers. The unit of the optical flow speed generally is pixels per image sequence (30min in this case). For our application we wanted to have sufficient contrast at low speeds, so the resulting optical flow speed in truncated at 1 and stored as 8-bit greyscale image with a maximum value of 255. This means a value 255 corresponds to an optical flow speed of 1 pixel/30 min, and a threshold value of 160 for MVIRI corresponds to a speed of 160/255 ~ 0.63 pixel/30 min. As the native pixel size of the visible channels for SEVIRI is larger, the speed threshold is reduced accordingly to 112/255 ~ 0.44 pixel/30 min. In other words, the threshold used for the optical flow speed is quite low in order to assure clouds are excluded from the snow detection process.  We will include/update the values in the revised manuscript.

Figure 3: Please include the definition of optical flow speed in the caption.

A: We will try to be clear concerning the optical flow speed in the revised manuscript.

Line 145-148. Are you saying that snow-coverage information is replaced by that of ERA5? Then why don't you use the ERA5 snow map and skip step 1 and 2?

A:  The snow-coverage information is not replaced, but corrected by ERA-5 in the way that snow is removed if it is not included in ERA-5. A reason for this correction is that there is a chance of misclassifying fog as snow, which is considered in the algorithm but not fully solvable.

Line 154. Albedo is a hemispherical variable. It is awkward that albedo is derived by the ratio of radiance differences.

A:  You are referring to the Effective Cloud Albedo here (CAL). Yes, the albedo is defined as integral of reflectances over all wavelengths and over the whole hemisphere. In our approach we assume the cloud reflectance to be isotropic. Different viewing geometries are considered by calculating the CAL based on the normalized reflection and by using the minimum reflectance per time slot (e.g. for the 13:00 UTC time slot) and per pixel.

CAL might also be expressed as the normalized difference of the all sky and clear sky reflection. Since several decade this quantity is called Effective Cloud Albedo.

Line 186 change radiation to irradiance.

A: Agreed. We will use the term irradiance here. This will be included in the updated manuscript.

Line 187 Need some explanations of how to get the relationship.

A: For the calculation of SID we would like to refer to Müller et al., 2015 and Skartveit, A., Olseth, J.A. and Tuft, M.A. (1998) An Hourly Diffuse Fraction Model with Correction for Variability and Surface Albedo. Solar Energy, 63, 173-183. We will point to these references for SID in the revised manuscript.

Line 300 Need brief description of Roesch et al's approach to derive monthly means.

A: We will add a brief description of the how the BSRN data is averaged to daily and monthly means. Roesch et al. proposes a monthly mean calculation method of 1-minute solar irradiance data that makes use of the diurnal cycle of solar irradiance. Hence the monthly mean is not disturbed from missing values.

BSRN and GEBA data: Are these stations listed in Tables 2 and 3 available for the entire period of 1983 to 2020? If they are not, please add column to indicate available time period for each station.

A: Thank you for the comment. GEBA data is available for the entire period of 1983 to 2020, but BSRN is not. We will add the temporal data coverage for the BSRN stations.

Figure 11: Why are absolute values used in these plots. I suggest using RMS differences.

A: The mean absolute difference (MAD) has been found to be very easy to understand and interpret by the users. Further the MAD is used in the CM SAF surface radiation data record validation as the main measure to define the quality of a data record. For these reasons we would like to stick to the absolute values. By using the boxplots the spread of the differences is nicely shown. However, we consider to add the RMS values for each station to the plots.

Figure 13: Are there any reasons for using sunshine duration, or not using global irradiances, in the figure. Figures 14 and 15 uses global irradiances. Please include global irradiance anomaly time series plot for the BSRN comparison.

A: **TO DO**: The reason for using sunshine duration in this plot is that there are much more reference measurements for sunshine durations and their temporal availability allows the analysis of the whole SARAH-3 climate data record. BSRN data starts in 1994 which is likely too late to see any effect of the Pinatubo eruption. Considering the small number of BSRN stations and their heterogeneous data coverage, an anomaly series over time would be difficult to interpret. Those issues diminish by using the monthly sunshine duration data from CLIMAT as reference in Figure 13. For the long-term stability analysis, the GEBA data is more appropriate due to its better spatial and longer temporal coverage, even though limited to Europe. Nevertheless, we will consider adding the BSRN bias time series to the revised manuscript.

Figures 13, 14, and 15. It is not clear whether a downward trend exists in parameterized irradiance, or observations, or both. Figure 15 suggests that trends for both computed and observed irradiances are mostly positive, but the trend of the difference is negative because observations have a large positive trend. Is this true for BSRN? Also, the negative trend is largely due to positive difference before 1995 for both BSRN and GEBA. Please investigate further why the difference is larger before 1995. In addition, if the number of surface sites changes over the course of the time period, it might introduce a trend.

A: You are right, Figure 13 and 14 show the bias time series of SARAH-3 vs station observations and do not show absolute trends. Both Figures 13 and 14 show that there is a small negative trend in the bias for sunshine duration (Figure 13) and for global irradiance. That means that a trend in the surface reference measurements would be underestimated by the SARAH-3 data record. Indeed, the overall negative trend is partly caused by the positive anomalies in the early years of the data record, which in turn might be caused by the underestimation of the direct aerosol effect during that period. The strong Pinatubo eruption seems to play a role here, too. For Figure 14 and 15 the number of stations available over time is constant to avoid artificial trends. Figure 15 shows the absolute trends of global irradiance. A constant number of stations over the whole time period has be used. The "Trendraster"-plots have been calculated and plotted for SARAH-3 and the GEBA surface measurements. Figure 15 reveals that both data sources see a positive trend in global irradiance of ~3.3 W/m2/decade for the GEBA statins and ~2.7 W/m2/decade for SARAH-3 global irradiance. Hence there is a negative trend in the bias (SARAH-3 minus GEBA) of ~-0.6 W/m2/decade. This value is also seen in Figure 14 (left) as number in lower part. The data basis for Figure 14 and 15 are similar.

Line 514 to 525. I assume that these discussions are for Figure 16. There are many lines in Figure 16 and hard to see. I suggest separating Figure 16 into two or more plots.

A: **TO DO** Yes, lines 514 to 525 discuss Figure 16. We will add the missing reference to Figure 16 here. Figure 16 is meant to qualitatively explain the relation between the SARAH-3 variables by showing zonal means. Splitting the Figure is an option but the relation between the variables would be less well visible. We suggest enlarging the figure and better streamline the discussion.

---

## Author Response (AR1)

**Authors Response to the Review of the ESSDD Manuscript "SARAH-3 – satellite-based climate data records of surface solar radiation"**

Authors: Uwe Pfeifroth, Jaqueline Drücke, Steffen Kothe, Jörg Trentmann, Marc Schröder, and Rainer Hollmann

Handling topic editor: Di Tian, tiandi@auburn.edu

Dear Ladies and Gentlemen,
Dear Referees,
Dear Editor,

Thanks a lot for the handling of the manuscript "ESSD-2024-91"; we have now updated the manuscript and uploaded the relevant documents. The revised manuscript has been carefully compiled by addressing all the referee comments from the public discussion.

Thanks to the reviewer comments, the manuscript was adjusted and improved. Additionally, we partly extended the manuscript, e.g. the list of references has been updated to include more recent relevant peer-reviewed articles. As, in the meantime, updates of the BSRN reference data were available, we have also updated the comparison of the satellite data with the BSRN data to cover the latest periods. The methods, and the overall results and conclusions of the original manuscript have not been altered.

Our comments and responses to the public discussion have been considered and integrated accordingly into the revised version of the manuscript. In the track-change version of the revised manuscript we refer to the corresponding referee comment for the revised part of the manuscript.
We would like thank again the two anonymous reviewers and Seiji Kato for their constructive review which strongly improved the manuscript.

Best regards,
Uwe Pfeifroth and coauthors